

# The Open Global Glacier Model (OGGM) v1.0

Fabien Maussion[1], Anton Butenko[2], Julia Eis[2], Kévin Fourteau[1,3], Alexander H. Jarosch[4],
Johannes Landmann[1,5,6], Felix Oesterle[1], Beatriz Recinos[2], Timo Rothenpieler[2], Anouk Vlug[7], Christian T. Wild[1,8], and Ben Marzeion[2]

[1]Department of Atmospheric and Cryospheric Sciences, Universität Innsbruck, Innsbruck, Austria
[2]Institute of Geography, University of Bremen, Bremen, Germany
[3]Univ. Grenoble Alpes, CNRS, IRD, Grenoble INP, IGE, 38000 Grenoble, France
[4]Institute of Earth Sciences, University of Iceland, Reykjavík, Iceland
[5]Laboratory of Hydraulics, Hydrology and Glaciology (VAW), Swiss Federal Institute of Technology (ETH), Zürich, Switzerland
[6]Swiss Federal Institute for Forest, Snow and Landscape Research (WSL), Birmensdorf, Switzerland
[7]MARUM - Center for Marine Environmental Sciences and Faculty of Geosciences, University of Bremen, Bremen, Germany
[8]Gateway Antarctica, Centre for Antarctic Studies and Research, University of Canterbury, Christchurch, New Zealand

*Correspondence to:* F. Maussion (fabien.maussion@uibk.ac.at)

**Abstract.**

Despite of their importance for sea-level rise, seasonal water availability, and as source of geohazards, mountain glaciers are one of the few remaining sub-systems of the global climate system for which no globally applicable, open source, community-driven model exists. Here we present the Open Global Glacier Model (OGGM, www.oggm.org), developed to provide a mod-

ular and open source numerical model framework for simulating past and future change of any glacier in the world. The modelling chain comprises data downloading tools (glacier outlines, topography, climate, validation data), a preprocessing module, a mass-balance model, a distributed ice thickness estimation model, and an ice flow model. The monthly mass-balance is obtained from gridded climate data and a temperature index melt model. To our knowledge, OGGM is the first global model explicitly simulating glacier dynamics: the model relies on the shallow ice approximation to compute the depth-integrated

flux of ice along multiple connected flowlines. In this paper, we describe and illustrate each processing step by applying the model to a selection of glaciers before running global simulations under idealized climate forcings. Even without an in-depth calibration, the model shows a very realistic behaviour. We are able to reproduce earlier estimates of global glacier volume by varying the ice dynamical parameters within a range of plausible values. At the same time, the increased complexity of OGGM compared to other prevalent global glacier models comes at a reasonable computational cost: several dozens of glaciers can be

simulated on a personal computer, while global simulations realized in a supercomputing environment take up to a few hours per century. Thanks to the modular framework, modules of various complexity can be added to the codebase, allowing to run new kinds of model intercomparisons in a controlled environment. Future developments will add new physical processes to the model as well as tools to calibrate the model in a more comprehensive way. OGGM spans a wide range of applications, from ice-climate interaction studies at millenial time scales to estimates of the contribution of glaciers to past and future sea-level

change. It has the potential to become a self-sustained, community driven model for global and regional glacier evolution.





## 1 Introduction

Glaciers constitute natural low-pass filters of atmospheric variability. They allow people to directly perceive slow changes of the climate system, that would otherwise be superimposed by short-term noise in human perception. Since glaciers form prominent features of many landscapes, shrinking glaciers have become an icon of climate change.

However, impacts of glacier change – whether growth or shrinkage – go far beyond this sentimental aspect: glaciers are important regulators of water availability in many regions of the world (Kaser et al., 2010; Huss, 2011; Immerzeel et al., 2012), and retreating glaciers can lead to increased geohazards (see Richardson and Reynolds, 2000, for an overview). Even though the ice mass stored in glaciers is small compared to the Greenland and Antarctic ice sheets ($< 1\%$), glacier melt has contributed significantly to past sea-level rise (SLR; e.g. Cogley, 2009; Leclercq et al., 2011; Marzeion et al., 2012b; Gardner et al., 2013).

They probably have been the biggest single source of observed SLR since 1900 and they will continue to be a major source of SLR in the 21st century (e.g. Church et al., 2013; Gregory et al., 2013).

    It is therefore a pressing task to improve the knowledge on how glaciers change when subjected to climate change, both natural and anthropogenic (Marzeion et al., 2014a). The main obstacle to achieve progress is a severe undersampling problem: direct glaciological measurements of mass balances have been performed on $\sim$300 glaciers world wide ($\approx 0.1\%$ of all glaciers

on Earth). The number of glaciers on which these types of measurements have been carried out for time periods longer than 30 years, i.e. over periods that potentially allow for the detection of a climate change signal, is one order of magnitude smaller (Zemp et al., 2009). Length variations of glaciers have been observed for substantially longer periods of time (Oerlemans, 1994, 2005). These variations are, however, much more difficult to understand, as large glacier length fluctuations may arise from intrinsic climate variability (Roe and O'Neal, 2009; Roe, 2011). Data obtained by remote sensing allow for gravimetric

assessments of ice mass change or volume change estimates obtained by differencing digital elevation models. Unfortunately, though, they are only available for the past decade (e.g. Gardner et al., 2013).

    During the past few years, great progress has been made in methods to model glaciers globally (Radić and Hock, 2011, 2014; Giesen and Oerlemans, 2012, 2013; Marzeion et al., 2012a, b, 2014a, b; Huss and Hock, 2015). While these approaches yield consistent results at the global scale, all of them suffer from greater uncertainties at the regional and local scales. These stem

from the great level of abstraction of the key processes (Marzeion et al., 2012b, 2014b), from the need to spatially interpolate model parameters (Radić and Hock, 2011, 2014; Giesen and Oerlemans, 2012, 2013), and from uncertainties of the boundary and initial conditions. All models lack ice dynamics, most (with the exception of Huss and Hock, 2015) lack calving, and all lack modulation of the surface mass balance by debris cover and snow redistribution (wind and avalanches). Only one model (Marzeion et al., 2012b) was able to provide estimates of past glacier volume changes for the 20th century. None of these

models is open-source.

    Mountain glaciers are one of the few remaining subsystems of the global climate system for which no globally applicable, open source, community-driven model exists. The ice sheet modelling community shows a better example, with models such as the *Parallel Ice Sheet Model* (Winkelmann et al., 2011) or *Elmer/Ice* (http://elmerice.elmerfem.org/). While the atmospheric modelling community has a long tradition of sharing models (e.g. the *Weather Research and Forecasting model*, or WRF) or





comparing them (e.g. the *Coupled Model Intercomparison Project* or CMIP), recent initiatives originating from the glacio-
logical community show a new willingness to better coordinate global research efforts following the CMIP example (e.g. the
*Glacier Model Intercomparison Project*[1] or the *Glacier Ice Thickness Estimation Working Group*[2]).

In the recent past, great advances have been made in the global availability of data and methods relevant for glacier modelling,
spanning glacier outlines (Pfeffer et al., 2014), automated glacier centerline identification (e.g., Kienholz et al., 2014), bed
rock inversion methods (e.g., Huss and Farinotti, 2012), and global topographic data sets (e.g. Farr et al., 2007). Taken together,
these advances now allow the ice dynamics of glaciers to be simulated by global scale models, provided that adequate modelling
platforms are available. In this paper, we present the **Open Global Glacier Model (OGGM)**, developed to provide a modular
and open source numerical model framework for consistently simulating past and future global scale glacier change.

*Global* not only in the sense of leading to meaningful results for all glaciers combined, but also for any small ensemble of
glaciers, e.g. at the headwater catchment scale. *Modular* to allow different approaches to the representation of ice flow and
surface mass balance to be combined and compared against each other. *Open source* so that the code can be read and used by
anyone and so that new modules can be added and discussed by the community, following the principles of open governance.
*Consistent* between past and future in order to provide uncertainty measures at all realisable scales.

This paper describes the basic structure and primordial assumptions of the model (as of version 1.0). We present the results
of a series of single glacier and global simulations demonstrating the model's usage and potential. This will be followed
by a description of the software requirements and the testing framework. Finally, we will discuss the potential for future
developments that could be conducted by any interested research team.

## 2  Fundamental principles

The starting point of OGGM is the Randolph Glacier Inventory (RGI; RGI Consortium, 2017; Pfeffer et al., 2014): our goal is
to simulate the past and future evolution of every single of the 216'502 inventoried glaciers worldwide (as of RGI V6). This
"glacier centric" approach is the one followed by most global and regional models to date; its advantages and disadvantages
will be discussed in Sect. 3.6.4. Provided with the glacier outlines, topographical and climate data at reasonable resolution and
accuracy, the model should be able to (i) provide a local map of the glacier including topography and hypsometry, (ii) estimate
the glacier's total ice volume and compute a map of the bedrock topography, (iii) compute the surface climatic mass balance
and (if applicable) at its front via calving, (iv) simulate the glacier's dynamical evolution under various climate forcings, and
(v) provide an estimate of the uncertainties associated with the modelling chain.

For each of these steps, several choices are possible regarding the input data to be used, the numerical solver or the parame-
terisations to be applied. Any given choice is driven by subjective considerations about data availability, the estimated accuracy
of boundary conditions (such as topography), and by technical considerations such as the available computational resources.
In this paper we present one way to realize these steps using OGGM, which to date is in our opinion the best compromise

---

[1]http://www.climate-cryosphere.org/activities/targeted/glaciermip
[2]http://www.cryosphericsciences.org/wg_glacierIceThickEst.html



between model complexity, data availability and computational effort. The OGGM software, however, is built in such a way that future improvements can be implemented, tested, and applied at minimal cost.

## 2.1 Example workflow

We illustrate with an example how the OGGM workflow is applied to the Tasman glacier, New Zealand (Fig. 1). Here we
describe shortly the purpose of each processing step, and more details will be provided in Sect. 3:

**Preprocessing** The glacier outlines extracted from the RGI are projected onto a local gridded map of the glacier (Fig. 1a). Depending on the glacier's location, a suitable source for the topographical data is downloaded automatically (here SRTM) and interpolated to the local grid. The map's spatial resolution depends on the size of the glacier (here, 150 m).

**Flowlines** The glacier centerlines are computed using a geometrical routing algorithm (adapted from Kienholz et al., 2014,
Fig. 1b), filtered and slightly modified to become glacier flowlines with a fixed grid spacing.

**Catchment areas and widths** The geometrical widths along the flowlines are obtained by intersecting the normals at each grid point with the glacier outlines and the tributaries' catchment areas. Each tributary and the main flowline has a catchment area, which is then used to correct the geometrical widths so that the flowline representation of the glacier is in close accordance with the actual altitude-area distribution of the glacier (Fig. 1d, note that the normals are now
corrected and centred).

**Climate data and mass balance** Gridded climate data (monthly temperature and precipitation) are interpolated to the glacier location and temperature is corrected for altitude using a linear gradient. These climate time series are used to compute the glacier mass balance at each flowline's grid point for any month in the past.

**Ice thickness inversion** Using the mass balance data computed above and relying on mass-conservation considerations, an
estimate of the ice flux along each glacier cross-section can be computed. By making assumptions about the shape of the cross-section (parabolic or rectangular) and using the physics of ice flow, the model computes the thickness of the glacier along the flowlines and the total volume of the glacier (Fig. 1e).

**Glacier evolution** A dynamical flowline model is used to simulate the advance and retreat of the glacier under preselected climate time series. Here (Fig. 1f), a 100-yr long random climate sequence leads to a glacier advance.

## 2.2 Model structure

The OGGM model is built around the notion of tasks, which have to be applied sequentially to single or a set of glaciers. There are two types of tasks:

**Entity tasks** are tasks which are applied on single glaciers individually and do not require information from other glaciers (this encompasses the majority of OGGM's tasks). Most often they need to be applied sequentially (for example, it is
not possible to compute the centerlines without having read the topographical data first).





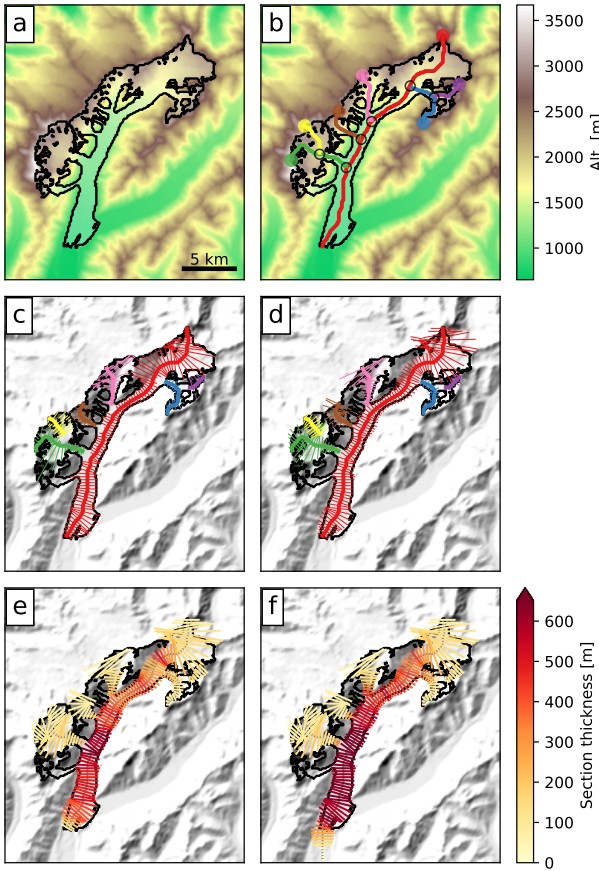

**Figure 1.** Example of the OGGM workflow applied to the Tasman glacier, New Zealand; **a**: topographical data preprocessing; **b**: computation of the flowlines; **c**: geometrical glacier widths determination; **d**: width correction according to catchment areas and altitude-area distribution; **e**: ice thickness inversion; **f**: random 100-yr long glacier evolution run leading to a glacier advance. See Sect. 2.1 for details.

**Global tasks** are tasks which are run on a set of glaciers. This encompasses the calibration and validation routines, which need to gather data across a number of reference glaciers.

This model structure has several advantages: the same entity task can be run in parallel on several glaciers at the same time, and they allow a modular workflow. Indeed, a task can seamlessly be replaced by another similar one, as long as the required

5    input and output formats are agreed upon beforehand. The output of each task is made persistent by storage on disk, allowing a later use by a subsequent task, even in a separate run or on another machine. For example, the preprocessing tasks store the topography data in a netCDF file, which is then read by the centerlines task, which itself writes it's output in a vector file format.





In this paper we will refrain from naming the tasks by their function name in the code, as these are likely to change in the future and are sometimes organised in a non-trivial way as a result of implementation details. The next section therefore is called "Modules", where each module can be seen as a collection of tasks developed towards a certain goal.

## 3 Modules

The modules are described in the order in which they are applied for a model run. When we provide a specific value for a model parameter in the text, we refer to the model's default parameter value: it can be changed at runtime by the user.

### 3.1 Preprocessing

The objective of the preprocessing module is to set up the geographical input data for each glacier (the glacier outlines and the local topography). First, a Cartesian local map projection is defined: we use a local Transverse Mercator projection centred on

the glacier. Then, a suitable topographical data source is chosen automatically, depending on the glacier's location. Currently we use:

- the Shuttle Radar Topography Mission (SRTM) 90m Digital Elevation Database v4.1 (Jarvis et al., 2008) for all locations in the [60°S; 60°N] range

- the Greenland Mapping Project (GIMP) Digital Elevation Model (Howat et al., 2014) for mountain glaciers in Greenland

(RGI region 05)

- the Radarsat Antarctic Mapping Project (RAMP) Digital Elevation Model, Version 2 (Liu et al., 2015) for mountain glaciers in Antarctica (RGI region 19 with the exception of some peripheral islands)

- the Viewfinder Panoramas DEM3 product (http://viewfinderpanoramas.org/dem3.html) elsewhere (most notably: North America, Russia, Iceland, Svalbard)

All datasets have a comparable spatial resolution (from 30 to 90 m, or 3 arcseconds). Using different data sources is problematic but unavoidable since there is no consistent and globally available Digital Elevation Model (DEM) to date. The Advanced Spaceborne Thermal Emission and Reflection Radiometer (ASTER) Global Digital Elevation Model Version 2 (GDEM V2) is available globally but was quickly eliminated because of large data voids and artefacts, in particular in the Arctic. These artefacts are often tagged as valid data and cannot be detected automatically in an easy way. The Viewfinder Panoramas products

instead have been corrected manually (mostly with topographic maps; J. de Ferranti, Pers. Comm.) and thus ensure a more realistic void filling. Although having a nearly global coverage, the DEM3 products are not used in place of established and citable digital elevation models such as SRTM, GIMP or RAMP, because of the lack traceability of the original data sources used to generate them. It must be noted that a number of glaciers will still suffer from poor topographic information. Either the errors are large or obvious (in which case the model won't run), or they are left unnoticed. The importance of reliable

topographic data for global glacier modelling will be the topic of a follow-up study.





The spatial resolution of the target grid depends on the size of the glacier: the default is to use a square relation to the glacier size ($dx = aS^2$ with $a = 14$ and $S$ the area of the glacier in km$^2$) clipped to a predefined minimum (10 m) and maximum (200 m) value. After the interpolation to the target grid, the topography is smoothed with a Gaussian filter of 250 m radius. This smoothing is driven by practical considerations, since the model becomes unstable if the boundary conditions are too
noisy (see also Bahr et al., 2014, for a discussion about the unavoidable trade-off between resolution and accuracy).

## 3.2   Flowlines and catchments

The glacier centerlines are computed following an algorithm developed by Kienholz et al. (2014) and adapted for our purposes. This algorithm was chosen because it allows to compute multiple centerlines and to define a main branch fed by any number of tributaries. In general we found the method to be very robust, although some glaciers obviously won't have the optimal number
of centerlines, with either too many (frequent in the case of large cirque glaciers) or not enough (some tributary branches have no centerlines). These errors however are assumed to play a relatively minor role in comparison to other uncertainties in the model chain.

In the model semantics, the original centerlines are then converted to flowlines: the points defining the line geometries are interpolated to be equidistant from each other (the default grid spacing is twice that of the map topography), and the tail of the
tributaries are cut before reaching their descendant (see the differences between Fig. 1b and c). Each grid point's elevation is obtained from the underlying topography. By construction, deepenings and upslopes along the flowline are very rare: this can still occur when the glacier outlines are poorly defined or when they do not match the gridded topography. In these cases, we interpolate the heights (in the case of a deepening) or cut the first grid points of the line (in case of an upslope starting from the flowline's head) until only positive slopes larger than 1.5° remain. This is necessary because the glacier's flowlines aren't
physically allowed to go up or have zero slope.

The flowlines are then sorted according to their Strahler number (a measure of branching complexity defined by Strahler, 1952, and commonly used in hydrological applications), from the lowest (line without tributaries but with possible descendants) to the highest (the main – and longest – centerline). This ordering is important for the mass flow routing: indeed, each flowline contains a reference to its descendant, and this reference is used by the inversion and dynamical models to transfer mass from
the tributaries towards the main flowline.

The width of each grid point along the flowline is computed in four steps. First, the catchment area of each flowline is computed using a routing algorithm similar to that used to compute the centerlines (Fig. 2a). Then the geometrical widths are computed by intersecting the flowline's normal to the boundaries of either the individual catchments or the glacier itself (Fig. 2b). These geometrical widths are then corrected by a factor specific for each altitudinal bin (Fig. 2c), so that the true
altitude area distribution of the glacier is approximately preserved (Fig. 2d). Finally, these widths are multiplied by a single factor ensuring that the total area of the glacier is exact same as the one provided by the RGI, ensuring consistency with future model intercomparisons.

At this stage, it is important to note that the map representation of the flowline glacier presented in Fig. 2c is purely artificial. The fact that the glacier cross-sections are overlapping is irrelevant: the role of the flowlines is to represent the actual flow of



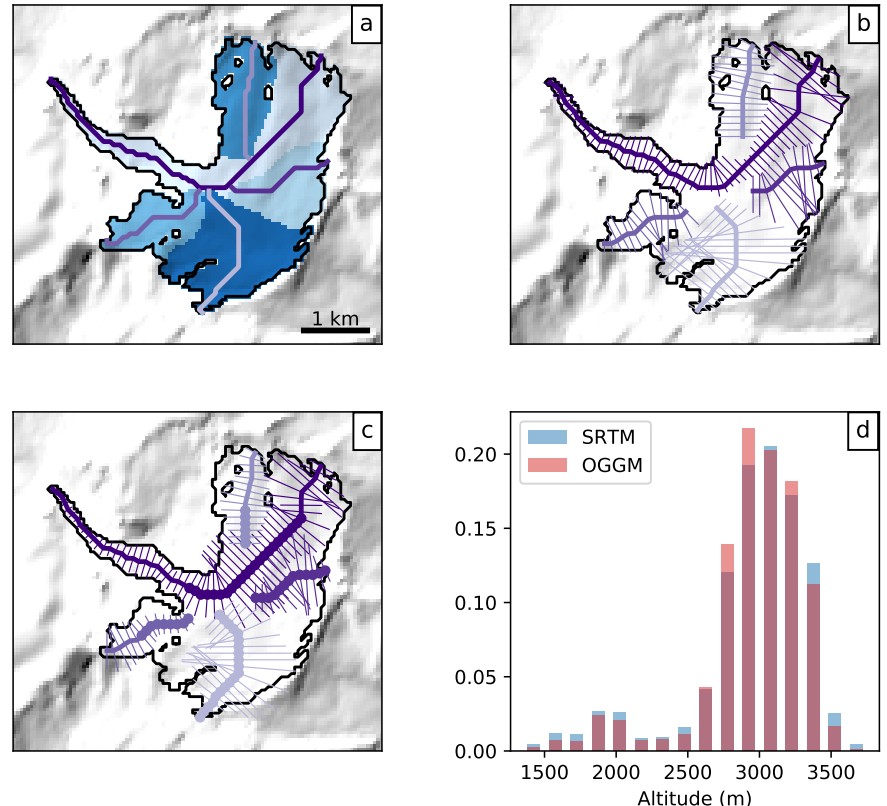

**Figure 2.** Example of the flowlines' width determination algorithm applied to the Upper Grindelwald glacier, Switzerland; **a**: determination of each flowline's catchment area; **b**: geometrical widths; **c**: widths corrected for the altitude-area distribution, the bold lines representing the grid points where the cross-section touches a neighbouring catchment; **d**: frequency distribution of the glacier area per altitude bin, as represented by OGGM and by the SRTM topography.

ice as accurately as possible while conserving the fundamental aspects of the real glacier: slope, altitude, area, geometry. The flowline approximation is going to work better for valley glaciers (like Tasman glacier shown above) than for cirque glaciers (like the Upper Grindelwald). For ice-caps, the flowline representation is likely to work poorly, as discussed in Sect. 3.6.2. From Fig. 2c one can see that future improvements of the mass balance model based on e.g. topographical shading or snow redistribution are made possible by the knowledge about the flowlines' location.



### 3.3 Climate data and mass balance

The mass balance model implemented in OGGM is an extended version of the temperature index melt model presented by Marzeion et al. (2012b). The monthly mass balance $m_i$ at an elevation $z$ is computed as:

$$m_i(z) = p_f \, P_i^{Solid}(z) - \mu^* \, max\left(T_i(z) - T_{Melt}, 0\right) \tag{1}$$

where $P_i^{Solid}$ is the monthly solid precipitation, $p_f$ a global precipitation correction factor (see Appendix A), $T_i$ the monthly air temperature and $T_{Melt}$ is the monthly air temperature above which ice melt is assumed to occur (default: -1°C, chosen because melting days can occur even if the monthly average temperature is below 0°C). Solid precipitation is computed as a fraction of the total precipitation: 100% solid if $T_i <= T_{Solid}$ (default: 0°C), 0% if $T_i >= T_{Liquid}$ (default: 2°C), and linearly interpolated in between. The parameter $\mu^*$ indicates the temperature sensitivity of the glacier and needs to be calibrated. For

this paper, the temperature and precipitation time series (1901–2016) are obtained from gridded observations (CRU ts4.01; Harris et al., 2014, see Appendix A). The temperature lapse-rate is set to a constant value (default: 6.5 K km$^{-1}$) or it can be time-dependant and computed from a linear fit of the 9 surrounding grid-points.

     For the calibration of the temperature sensitivity parameter $\mu^*$ we use the method described by Marzeion et al. (2012b) and successfully applied many times since then (e.g. Marzeion et al., 2014a, 2015). Although the general procedure didn't change,

its peculiarity justifies to spend some time describing it here. We will start by noting that $\mu^*$ depends on many factors, most of them being glacier-specific (e.g. avalanches, topographical shading, cloudiness), and others being related to systematic biases in the input data (e.g. climate, topography). As a result, $\mu^*$ can vary greatly between neighbouring glaciers without obvious physical reasons. The calibration procedure implemented in OGGM makes use of these apparent handicaps by turning them into assets.

The procedure begins with glaciers for which we have direct observations of specific mass balance ($N$ = 254, see Appendix B). For each of these glaciers, annual sensitivities $\mu(t)$ are computed from Eq. 1 by requiring that the glacier specific mass balance $\overline{m}(t)$ is equal to zero[3]. $\overline{m}(t)$ is the glacier integrated mass balance computed for a 31 yr period centred around the year $t$ and *for a constant glacier geometry fixed at the RGI outline's date* (e.g. 2003 in the Alps). The process is illustrated in Fig. 3c (blue line): around 1920 the climate was cold and wet (Figs. 3a and b), and as a consequence the hypothetical

temperature sensitivity required to maintain the 2003 glacier geometry needs to be high. Inversely, the more recent climate is warmer and the temperature sensitivity needs to become smaller for the glacier to remain stable.

     These hypothetical, time-dependent $\mu(t)$ are called "candidates": it is likely (but not certain) that at least one of them is the correct $\mu^*$. To determinate which of the candidates is suitable, we then compute the mass balance time series for each of the $\mu(t)$ and compute their bias with respect to observations. Note that the period over which the observations are taken is not

relevant for the bias computation: each $\mu$ candidate can produce a mass balance for any year, as per Eq. 1. This bias is shown in Fig. 3c (red line): in comparison to observations, $\mu(t = 2000)$ is too low and produces mass balances with a positive bias.

---

[3]Note that this is not valid for tidewater glaciers, where mass loss happens at the tongue and the equilibrium surface mass-balance budget doesn't have to be closed. See Sect. 3.6.1 for more details.



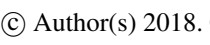

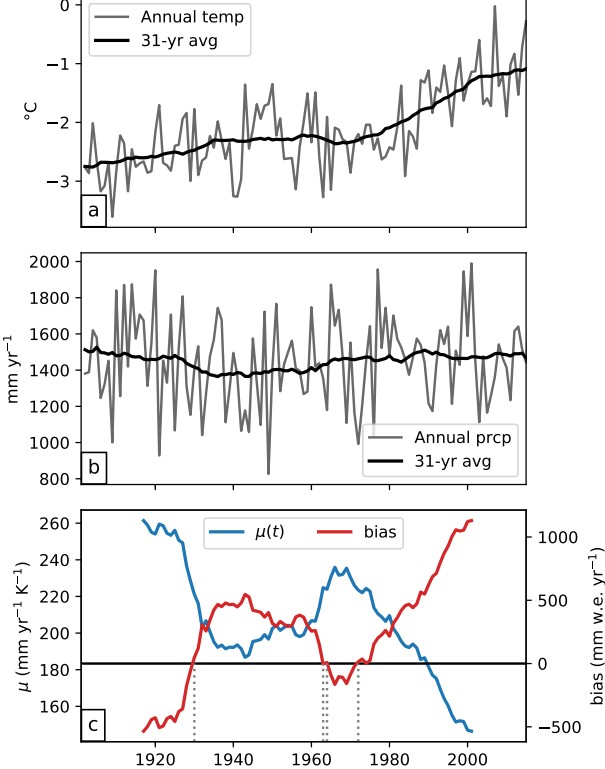

**Figure 3.** Calibration procedure for $\mu^*$ applied to the Hintereisferner glacier, Austria. **a** and **b**: annual and 31-yr average of temperature and precipitation obtained from the nearest CRU grid point (altitude 2700 m a.s.l.). **c**: time series of the $\mu$ candidates (mm yr$^{-1}$ K$^{-1}$) and their associated mass balance bias (mm w.e. yr$^{-1}$, right-axis) in comparison to observations. The vertical dashed lines mark the times where the bias is closest to zero.

Inversely, $\mu(t = 1920)$ is too high and leads to a negative bias. For four years, the bias is close to or crossing the zero line and $\mu(t)$ is therefore very close to the ideal $\mu^*$. These dates are called $t^*$, and represent the center of a 31-yr long climate period where *today's* glacier would be in equilibrium and maintain its *current* geometry. This $t^*$ is an actual date but is also an abstract concept: we are going to make use of it for the next step.

5   For the vast majority of the glaciers, $\mu^*$ and $t^*$ are unknown. For these we could interpolate the $\mu^*$ (maybe the most obvious solution), or we could interpolate $t^*$: indeed, the procedure above can be reversed and $t^*$ can be used to retrieve $\mu^*$, again by requiring that $\overline{m}(t^*)$ is equal to zero (Eq. 1). We interpolate $t^*$ to all glaciers without observations using inverse distance interpolation from the 10 closest locations. The residual bias for glaciers with observations can be close to zero (the case for Hintereisferner, where the bias curve crosses the zero line) but can also be higher (indicating that no 31-yr period in the last century would sustain the current glacier geometry). When no perfect $t^*$ is found, the date with the smallest absolute bias is chosen. This residual bias is also interpolated between locations and subtracted from the modelled mass balance. The benefit



of this approach is best shown by cross-validation (Fig. 4), where one can see that the error increases considerably when using $\mu^*$ interpolation instead of the proposed method. This is due to several factors:

- the equilibrium constraint applied on $\mu(t)$ implies that the sensitivity cannot vary much during the last century. In fact, $\mu(t)$ at one glacier often varies less in one century than between neighbouring glaciers, because of the local driving factors mentioned earlier. In particular, it will vary comparatively little around a given year $t$: errors in $t^*$ (even large) will result in relatively small errors in $\mu^*$.

- the equilibrium constraint will also imply that systematic biases in temperature and precipitation (no matter how large) will automatically be compensated by all $\mu(t)$, and therefore also by $\mu^*$. In that sense, the calibration procedure can be seen as an empirically driven downscaling strategy: if a glacier is located there, then the local climate (or the glacier temperature sensitivity) *must* allow a glacier to be there. For example, the effect of avalanches or a negative bias in precipitation input will have the same impact on calibration: the value of $\mu^*$ should be lowered to take these effects into account, even though they are not resolved by the mass balance model.

The most important drawback of this calibration method is that it assumes that two neighbouring glaciers should have a similar $t^*$. This is not necessarily the case, as other factors than climate (such as the glacier size) will influence $t^*$ too. Our results (and the arguments listed above) show however that this is an approximation we can cope with.

In a final note, it is important to mention that $\mu^*$ and $t^*$ should not be over-interpreted in terms of real temperature sensitivity or response time of the glacier. This procedure is primarily a calibration method, and as such it can be statistically scrutinized (for example with cross-validation). It can also be noted that the mass balance observations play a relatively minor role in the calibration: they could be entirely avoided by fixing a $t^*$ for all glaciers in a region (or even worldwide), without much performance loss. The observations, however, play a major role for the assessment of model uncertainty (Fig. 4). For more information about the climate data and the calibration procedure, refer to Appendix A.

### 3.4 Ice thickness

Measuring ice thickness is a labour-intensive and complex task, therefore only a fraction of the world's glaciers is monitored and direct measurements are sparse. A physical or statistical approach is necessary for modelling glacier evolution at the global scale. For a recent review of available techniques for ice thickness modelling, see Farinotti et al. (2017). OGGM implements a new ice thickness inversion procedure, physically consistent with the flowline representation of glaciers and taking advantage of the mass balance calibration procedure presented in the previous section. It is a mass-conservation approach largely inspired by Farinotti et al. (2009), but with distinct characteristics.

The principle is quite simple. The flux of ice $q$ [m$^3$ s$^{-1}$] through a glacier flux-gate (cross-section) of area $S$ [m$^2$] reads:

$$q = uS \tag{2}$$

with $u$ the average velocity [m s$^{-1}$]. Using an estimate for $u$ and $q$ obtained from the physics of ice flow and the mass balance field, $S$ and the local ice thickness $h$ [m] can be computed relying on some assumptions about the bed geometry. We





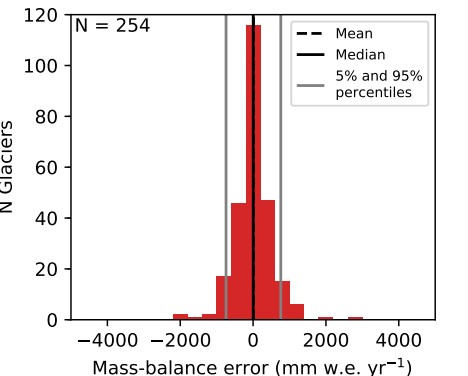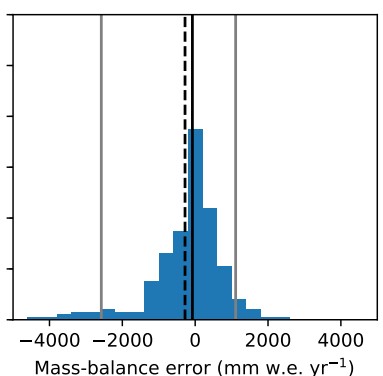

**Figure 4.** Benefit of spatially interpolating $t^*$ instead of $\mu^*$ as shown by leave-one-out cross-validation ($N = 254$). **Left**: error distribution of the computed mass balance if determined by the interpolated $t^*$. **Right**: error distribution of the mass balance if determined by interpolation of $\mu^*$. The vertical lines indicate the mean, median, 5% and 95% percentiles

compute the depth-integrated ice velocity using the well known shallow-ice approximation (Hutter, 1981, 1983):

$$u = \frac{2A}{n+2} h\tau^n \tag{3}$$

with $A$ the ice creep parameter [$\mathrm{s^{-1}\ Pa^{-3}}$], $n$ the exponent of Glen's flow law ($n$=3), and $\tau$ the basal shear stress, computed as:

$$\tau = \rho g h \alpha \tag{4}$$

with $\rho$ the ice density ($900\ \mathrm{kg\ m^{-3}}$), $g$ the gravitational acceleration ($9.81\ \mathrm{m\ s^{-2}}$) and $\alpha$ the surface slope computed numerically along the flowline. Optionally, a sliding velocity $u_s$ can be added to the deformation velocity to account for basal sliding. We use the same parameterisation as Oerlemans (1997), who relied on Budd et al. (1979):

$$u_s = \frac{f_s \tau^n}{h} \tag{5}$$

with $f_s$ a sliding parameter (default: $5{,}7 \times 10^{-20}\ \mathrm{s^{-1}\ Pa^{-3}}$). If we consider a point on the flowline and the catchment area $\Omega$ upstream of this point, mass conservation implies:

$$q = \int_\Omega \left(\dot{m} - \rho\frac{\partial h}{\partial t}\right) dA = \int_\Omega \widetilde{m}\, dA \tag{6}$$

with $\dot{m}$ the mass balance [$\mathrm{kg\ m^{-2}\ s^{-1}}$], and $\widetilde{m} = \dot{m} - \rho\partial h/\partial t$ the "apparent mass balance" after Farinotti et al. (2009). If the glacier is in steady state, the apparent mass balance is equivalent to the actual (and observable) mass balance. In the non-steady state case, $\partial h/\partial t$ is unknown, and neither is the time integrated (and delayed) mass balance $\int_\Omega \dot{m}$ responsible for the flux of ice through a section of the glacier at a certain time. Farinotti et al. (2009) and Huss and Farinotti (2012) deal with the issue





by prescribing an apparent mass balance profile as a parameterized linear gradient which is, arguably, more a semantic than a physical way to deal with the transience of the problem.

Like Huss and Farinotti (2012), OGGM cannot deal with the transient problem yet: we deliberately assume steady state and therefore set $\widetilde{m} = \dot{m}$. This has the strong advantage that we can make direct use of the equilibrium mass balance $\overline{m}(t^*)$

computed earlier, which satisfies $\int \overline{m} = 0$ by construction. $q$ is then obtained by integrating the equilibrium mass balance $\overline{m}$ along the flowline(s). The tributaries will have a positive flux at their last grid point: this mass surplus is then transferred to the downstream line, normally distributed around the 9 grid points centred at the flowlines' junction. By construction, $q$ starts at zero and increases along the major flowline, reaches its maximum at the equilibrium line altitude (ELA) and decreases towards zero at the tongue (for non-calving glaciers).

Equation 2 turns out to be a polynomial of degree 5 in $h$ with only one root in $\mathbb{R}_+$, easily computable for each grid point. The equation varies of a factor of 2/3 if one assumes a parabolic ($S = \frac{2}{3}hw$, with $w$ the glacier width) or rectangular ($S = hw$) bed shape. Shape factors as parameterisation for lateral bed stresses (Cuffey and Paterson, 2010) are currently not considered in OGGM, but it is in our short term plans to implement them. The default in OGGM is to use a parabolic bed shape, unless the section touches a neighbouring catchment (see Fig. 2c), neighbouring glacier (ice divides, computed from the RGI), or at

the terminus of a tidewater glacier. In theses cases the bed shape is rectangular. Singularities with flat areas are avoided since the constructed flowlines are not allowed to have a local slope $\alpha$ below a certain threshold (default: 1.5°, see Sect. 3.2).

Figure 5 displays some examples taken from the OGGM test suite, where the automated inversion procedure is applied on idealized glaciers generated with OGGM's flowline model (see Sect. 3.5). In the equilibrium cases (Fig. 5a to c), the inverted topography is nearly perfect. Differences arise at strong surface gradients, mostly because of numerical differences

(the inversion method uses a second order central difference which tends to smooth the slope). The transient case (Fig. 5d) illustrates the consequences of the steady-state assumption: although the glacier is shrinking, the constraint $\int \widetilde{m} = 0$ leads to a lowered ELA and, even with a perfectly known mass balance gradient, leads to an overestimated ice thickness. This effect is visible everywhere, but is strongest at the tongue.

The sensitivity of the inversion procedure to various parameters is illustrated with the example of the Hintereisferner glacier

(Fig. 6). The total volume (and the local thickness) is very sensitive to the choice of the creep parameter A, varied from a factor 1/10 to 10 times the default value of $2.4 \times 10^{-24}$ s$^{-1}$ Pa$^{-3}$ (Cuffey and Paterson, 2010). With a smaller A, the ice is stiffer and the glacier gets thicker (A is expected to get smaller by one or more orders of magnitude with colder ice temperatures). Inversely, softer ice leads to a thinner glacier. The shape of the curve is proportional to the fifth root of the fraction $1/A$, explaining why the volume gets very sensitive to small values of A. Adding sliding reduces the original thickness significantly

for the same reasons as an increasing A, since both sliding and ice rheology (A) have a strong influence on the computed ice flux $q$. The total precipitation amount, by acting on the mass balance gradient and therefore on the ice flux $q$ will also play a non-negligible role for the ice thickness (Fig. 6b). The effect is small in comparison to the influence of $A$, but it is consequent: glaciers located in maritime climates (with high values of accumulation) will be thicker on average than similar glaciers in drier conditions.



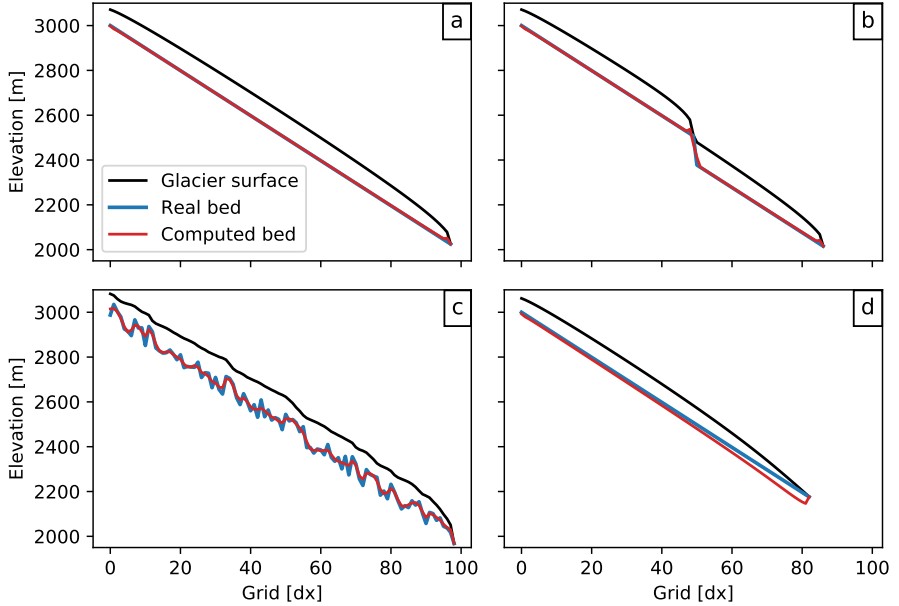

**Figure 5.** Idealized inversion experiments: we compute the bed topography out of the surface elevation obtained from a flowline model applied to a predefined bed topography. (**a**), (**b**) and (**c**): glacier grown to equilibrium with different bed topographies (flat, cliff, random). (**d**): transient experiment with a shrinking glacier. The same mass balance profile is used for all experiments (linear gradient of 3 mm w.e. m$^{-1}$, ELA altitude of 2600 m a.s.l.). For (d), the glacier is first grown to equilibrium then shrunk for 60 years after an ELA shift of +200 m.

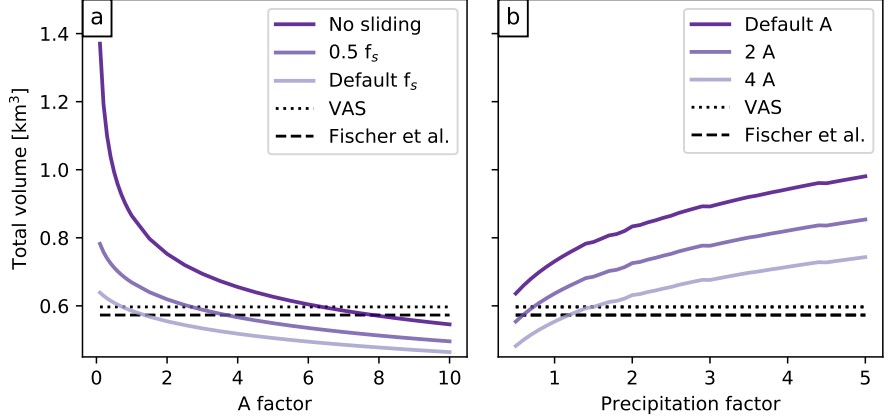

**Figure 6.** Total volume of the Hintereisferner glacier computed with varying factors for the default creep parameter A (**a**) and varying precipitation factor (**b**). The dotted and dashed lines display the total volume estimated with volume-area scaling (VAS, Bahr et al., 1997, 2015) and based on point observations (Fischer and Kuhn, 2013). For (**a**), additional sensitivities are computed with an additional sliding velocity (Oerlemans, 1997) and his sliding parameter f$_s$. For (**b**), additional sensitivities are computed with a varying creep parameter A.



This example shows that one can always find an optimum set of parameters leading to the correct total volume. In practice, however, calibrating the model for accurate global glacier volume estimates is a major challenge for global glaciological models and will be the topic of a separate study. The IACS Working Group on Glacier Ice Thickness Estimation[4] is working towards this goal: OGGM participated in the first Ice Thickness Models Intercomparison eXperiment (ITMIX, Farinotti et al., 2017),

ranking amongst the best models with limited data requirements.

### 3.5   Ice dynamics

At this stage of the processing workflow, the ice-dynamics module is straightforward to implement. Provided with the mass balance, slope, width $w$ and bed topography along the flowline, we solve the equation:

$$\frac{\partial S}{\partial t} = w\dot{m} - \nabla \cdot uS \tag{7}$$

numerically with a forward finite difference approximation scheme on a staggered grid. Numerical stability is ensured by the use of an adaptive time stepping scheme following the Courant-Friedrichs-Lewy (CFL) condition $\Delta t = \gamma \frac{\Delta x}{max(u)}$ with $\gamma$ as the dimensionless Courant number chosen between zero and one. Unlike many solvers of the shallow-ice equation, we do not transform Eq. 7 to become a diffusivity equation in $h$, but solve it as it is formulated here. This has the advantage that the numerical solver is the same regardless of the shape of the bed (parabolic, trapezoidal or rectangular). The new section $S$ at

time $t + \Delta t$ then translates in a certain $h$ according to the local bed geometry, allowing to have various bed geometries along the same flowline. The drawback of our approach is that we cannot take advantage of the diffusivity equation solvers already available elsewhere. We tested our solution against the robust and and mass-conservative solver presented by Jarosch et al. (2013): our model yields accurate (and faster) results in most cases, but fails to ensure mass-conservation for very steep slopes like most other solvers to date. While a flowline version of the solver presented by Jarosch et al. (2013) is available in OGGM,

it is not used operationally since it cannot yet handle varying bed shapes and multiple flowlines – it will become the default solver when these elements are implemented.

At a junction between a tributary and its downstream line, an artificial grid point is added to the tributary line. This grid point has the same section area $S$ and thickness $h$ as the previous one, but the surface slope is computed from the difference in elevation between tributary and descendant flowline. This is necessary to ensure a dynamical connection between the two

lines: when the main flowline is at a higher elevation than its tributary, no mass exchange occurs and the tributary will build up mass until enough ice is available. At a junction point, Eq. 7 therefore contains an additional mass flux term from the tributary.

Before the actual run, a final task merges the output of all preprocessing steps and initialises the flowline glacier for the model. For the glaciers to be allowed to grow, a downstream flowline is computed using a least cost routing algorithm leading the glacier towards the domain boundaries (this algorithm is similar to the algorithm used to compute the glacier centerlines).

The bed geometries along the downstream line are computed by fitting a parabola to the actual topography profile. In case of bad fit, the values are interpolated or a default parabola is used. Along the glacier, where the bed geometries are unknown

---

[4]http://www.cryosphericsciences.org/wg_glacierIceThickEst.html





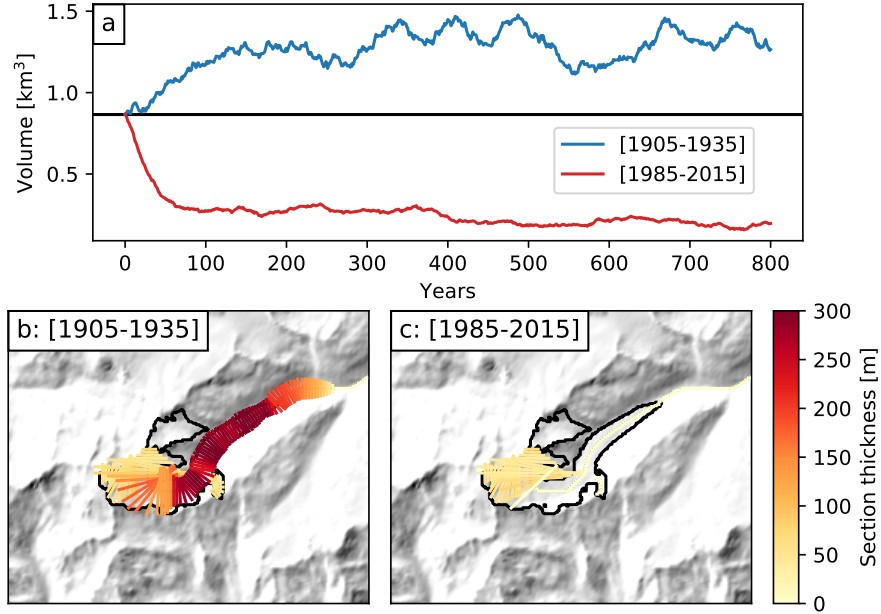

**Figure 7.** Evolution of the Hintereisferner under two random forcing scenarios and for the default parameter set. For each scenario, the "climate years" during a 31-yr period are shuffled randomly, therefore creating a realistic climate representative for a given period. (**a**): the glacier volume evolution for each scenario (the black line marks the initial computed glacier volume). (**b**) and (**c**): the glacier shape at the end of the 800 yrs simulation for each case.

before the inversion, the bed geometries are either rectangular (ice divides and junctions) or parabolic. Very flat parabolic shapes can happen occasionally, for wide sections with a shallow ice thickness. These geometries are unrealistically sensitive to changes in $h$. They create a strong positive feedback (the thickening of ice leading to a highly widening glacier) and are therefore prevented: when the parabola parameter falls below a certain threshold, the geometry is assumed to be trapezoidal

5     instead.

The results of two idealized simulations with a growing and a shrinking scenario are shown in Fig. 7. When put under the cold and wet climate of the beginning of the 20th century, Hintereisferner would grow about 2/3 larger than it is today. Inversely, the glacier is in strong disequilibrium with today's climate: it would lose about 2/3 of it's volume if the climate remained as it was during the past 31 yrs. The response time of the glacier is approximately twice as fast in the shrinking

10     case, and the natural random variations of the glacier are much smaller than for a large glacier with more inertia and a longer response time.

The previous results were obtained with the default set-up of OGGM. In Fig. 8 we assess the sensitivity of the dynamical model to changes in the creep parameters $A$ and to the addition of a basal sliding velocity. As expected, these dynamical parameters affect the equilibrium volume and the response time of the glacier (faster ice leading to a thinner glacier, and





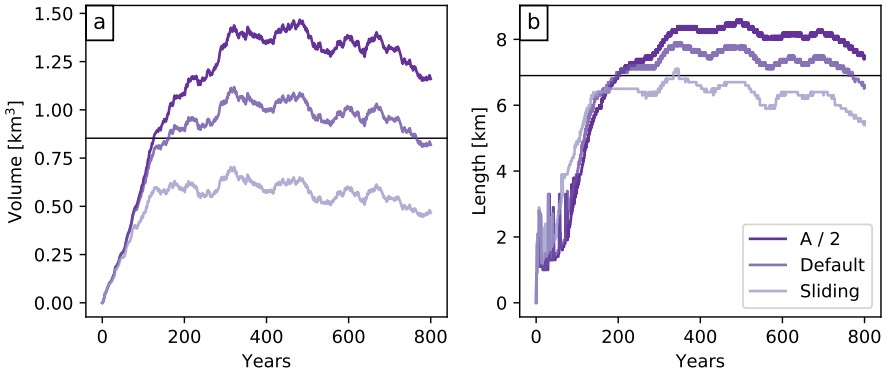

**Figure 8.** Evolution of volume (a) and length (b) of the Hintereisferner glacier under a random climate forcing generated by shuffling the "climate years" representative for the 31-yr period centred on $t^*$. The glacier is reset to zero for each simulation, and the bed topography is obtained with the default parameters. The sensitivity to the addition of a sliding velocity or to a halving of the creep parameter A are also shown. The noisy patterns of the length time series are due to the fact that the length of a glacier on a discrete grid is sensitive to small interannual variations.

inversely). Because of the mass-balance elevation feedback, the stiffer (low $A$) and therefore thicker glacier is also larger and longer, but its response to climate variability is smaller in amplitude than that of the fast moving sliding glacier.

$A$ and $f_s$ depend on many factors such as ice temperature or basal characteristics and they cannot be assumed to be globally constant. There are considered as calibration parameters in OGGM, and will be tuned towards observations of e.g. ice thickness

5 or glacier length changes. In this study we only calibrate the mass-balance model while the ice dynamics parameters are set to their default values (A = $2.4 \times 10^{-24}$ s$^{-1}$ Pa$^{-3}$ and $f_s = 0$). Nevertheless, we discuss the model sensitivity to these dynamical parameters for individual glaciers (Fig. 8) or global runs (Fig. 10).

### 3.6 Special cases and model limitations

The previous experiments demonstrate that the OGGM model is capable of simulating the dynamics of glaciers in a fully

10 automated manner. In this section we describe the implications of the flowline approximation in the special cases of tidewater glaciers and ice caps, and discuss some examples of glaciers with a less trivial geometry.

### 3.6.1 Tidewater glaciers

Glaciers are defined as tidewater in OGGM when their RGI terminus attribute is either flagged as marine-terminating or lake-terminating. The major difference between a tidewater glacier and a valley glacier is that mass-loss is occurring at the glacier

15 front (calving). This has implications for the bed thickness inversion which currently assumes that the mass-flux at the front is zero (by setting $\int \widetilde{m} = 0$, see Sect. 3.4) and for the dynamics of the glacier. The current treatment of tidewater glaciers in OGGM is very simple but explicit: we do not take calving into account for the bed inversion (i.e. the original calving front has



a thickness of zero), but we do have a basic calving parametrisation in the ice dynamics module. We add a grid point behind the calving front which is reset to zero ice thickness at each time step: the ice mass suppressed this way is the calving flux, that we store. This parametrisation has the advantage to prevent the tidewater glaciers to advance while still allowing them to retreat (in which case they stop to calve). We are currently working on a more advanced calving parametrisation for both the

ice dynamics and the ice thickness inversion, which will be the topic of a follow-up study.

### 3.6.2   Ice caps

Ice caps in the RGI are divided in single dynamical entities separated by their ice divide (Fig. 9). Currently, the only special treatment for ice caps in OGGM is that only the major flowline is computed without tributaries. Indeed, the geometry of ice caps is often non trivial and it is not clear whether tributaries would really improve the model results. An example of an ice

cap is shown in Fig. 9: while the general behaviour of the ice cap is reasonably simulated by the flowline model (e.g. at the outlet glaciers), other features appear to be unrealistic (e.g. close to the ice-divides). Moreover, the mass-conservation inversion method is probably underestimating the real ice-thickness at the location of the ice-divide, where other processes related to the past history of the ice cap are at play. A possible way forward would be to run a distributed shallow-ice model instead of the flowline representation, and it is part of our long-terms plans to do so.

### 3.6.3   Glacier complexes

Single glaciers can be defined as the smallest dynamically independent entity, i.e. the boundaries between two glaciers should approximately follow the ice divides or hydrological basin boundaries. The flowline assumption strongly relies on this condition being true, and indeed most of the RGI glaciers are properly outlined. Unfortunately there are notable exceptions, for three main reasons:

– human decision: some well known glaciers have historical boundaries that the inventory provider wanted to keep, although the glacier is now disintegrated in smaller entities. A good example is the Hintereisferner glacier (Fig. 7), which should have three outlines instead of one.

    – uncertainties in the topography: the inventories are often generated using both automated processes and manual editing. There is no guarantee that we use the same DEM as the original inventory, and therefore OGGM and RGI might disagree

on the ideal position of an ice divide.

    – unavailable data: some remote glaciers and ice caps are outlined in the RGI, but not divided at all. These are the most problematic cases, and should be a matter of concern for all RGI users. For example, the largest glacier in RGI (an ice cap in north-eastern Greenland with id RGI60-05.10315 and area 7537 km$^2$) is wrongly outlined and should be separated in at least a dozen of smaller entities.

Most of the small errors are filtered out by OGGM with algorithms based on surface slope thresholds (see Sect. 3.2), but the latter group of glaciers should be handled upstream. We have developed an open source tool to automatically compute glacier





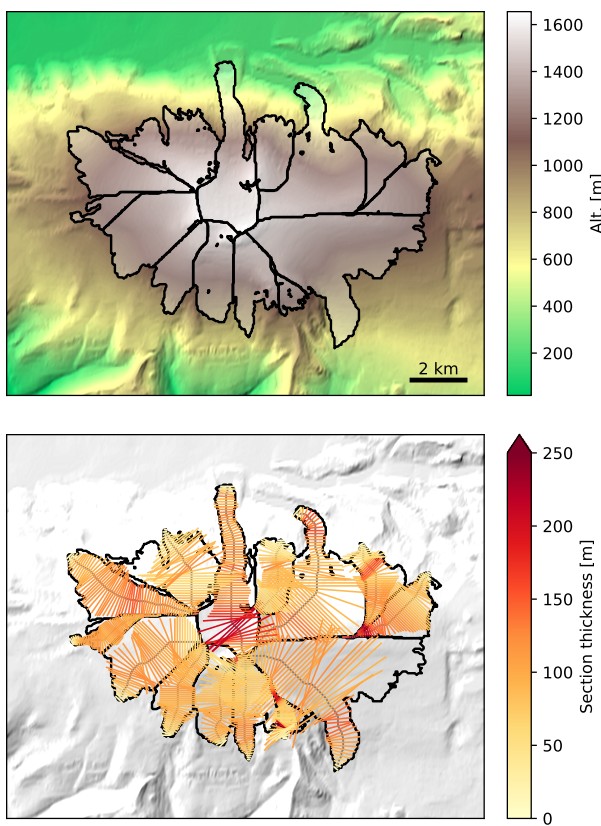

**Figure 9.** OGGM inversion workflow applied to the RGI entities of the Eyjafjallajökull ice cap, Iceland. **Upper panel**: outlines and topography. **Lower panel**: glacier thickness.

divides (https://github.com/OGGM/partitioning, based on Kienholz et al., 2013), but do not use it here. This issue is a large source of uncertainty for ice thickness estimates and dynamical modelling of glaciers in general, and will be the subject of a dedicated study.

### 3.6.4 Glacier centric modelling

5  Like most global glacier models, OGGM simulates each glacier individually. This has evident practical advantages, and is also a strong asset for our mass balance model calibration algorithm. However, this has two major drawbacks: (i) neighbouring glaciers won't merge although they grow together, and (ii) we can only simulate glaciers which are already inventoried, while uncharted glaciers are simply ignored. Both errors are a source of uncertainty for long or past simulations but less for short term projections in a warming world. The most obvious way to deal with this issue is to use distributed models (e.g. Clarke
10  et al., 2015), with their own drawbacks (mostly: computational costs and the need for distributed mass balance fields). Another way would be to allow the dynamical merging of neighbour flowline glaciers at run time. While both are viable options for

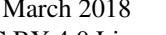

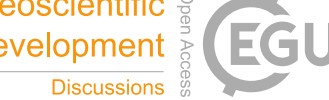


the OGGM workflow, they represent a considerable increase in complexity and are not available yet. Like other fundamental issues described in this paper (such as missing topographical data or wrongly outlined glaciers), this problem will affect other glacier models as well. We hope that some of the tools we introduce here will help to solve some of these issues upstream, and that the community will soon be able to put pressure on commercial data providers for better data availability.

## 4    Global simulations

Thanks to its automated workflow, OGGM is able to apply all processes described in the previous chapter to all glaciers of the globe with the exception of Antarctica, where no CRU data is available (see Appendix C for an overview of the RGI regions). No special model setup is needed, we use all model default settings without any calibration (this is not strictly true for the $\mu^*$ calibration, which is an automated process and cannot be tuned or turned off). In the following analyses the focus is put on the model behaviour and not on the quantitative results. However, as we are going to see our results are close to expectations even without calibration, indicating a realistic model behaviour.

### 4.1    Hardware requirements and performance

Thanks to the computational efficiency of the flowline model, OGGM runs quickly enough to be used on a personal computer for up to a hundred glaciers. At the global scale a high performance computing environment is required. For these global simulations we used a small-sized cluster comprising two nodes with 16 quad-core processors each, resulting in 128 parallel threads. With this configuration, the model preprocessing chain (including the ice thickness inversion) takes about 7 hours to complete (without data download). The total size of the preprocessed output is 170G, an amount which can be reduced by deleting intermediate computing steps. The amount of required storage increases with each dynamical run; here again it is possible to reduce the data amounts by storing only diagnostic variables such as volume, area, length, ELA instead of the full model output. The dynamical runs are the most expensive computations: running six 300-yr long global runs takes about 48 hours on our small cluster, a very satisfying performance. It is interesting to note that because of the adaptive time-step, glacier shrinkage scenarios run faster than growing ones.

### 4.2    Invalid glaciers

Due to uncertainties in the input data (topography, outlines, climate), a certain number of glaciers fail to be modelled by OGGM. The statistics of these invalid glaciers are summarized in Table 1. The largest amount of errors (2% of the total area) is due to invalid climate series. Errors occur mostly when the climate is too cold for melt to happen or, inversely, too warm or too dry for accumulation to happen. While some of these errors are directly due to incorrect climate data, some can also be attributed to missing processes in the OGGM mass balance model: sublimation and calving, both leading to mass-loss even at cold temperatures. The least problematic source of error (0.1% of the total area) is due to failures during the actual dynamical run. The large majority of dynamical failures (685 out of 710 glaciers) happen because the glacier exceeded the domain boundaries at run time. Some of these errors could be mitigated by increasing the domain size (at the cost of computational





**Table 1.** Statistics of the model errors for each RGI region. The column names indicate which processing step produces an error, the value is the number of invalid glaciers and (in parentheses) the percentage of regional area they represent.

|  | N | Area (km2) | Climate | Dynamics | Others | All |
|---|---|---|---|---|---|---|
| 01: Alaska | 27108 | 86725 | 57 (0.1%) | 2 (0.0%) | 20 (0.2%) | 79 (0.2%) |
| 02: Western Canada and US | 18855 | 14524 | 1 (0.0%) | 4 (0.0%) | 62 (0.5%) | 67 (0.5%) |
| 03: Arctic Canada North | 4556 | 105111 | 7 (0.0%) | 1 (0.0%) | 20 (0.1%) | 28 (0.1%) |
| 04: Arctic Canada South | 7415 | 40888 | 10 (0.0%) | 10 (0.0%) | 14 (0.2%) | 34 (0.2%) |
| 05: Greenland | 20261 | 130071 | 2186 (9.3%) | 480 (0.3%) | 133 (2.2%) | 2799 (11.8%) |
| 06: Iceland | 568 | 11060 |  |  | 1 (0.0%) | 1 (0.0%) |
| 07: Svalbard | 1615 | 33959 |  | 1 (0.0%) | 9 (2.9%) | 10 (2.9%) |
| 08: Scandinavia | 3417 | 2949 |  | 10 (0.1%) | 5 (0.1%) | 15 (0.1%) |
| 09: Russian Arctic | 1069 | 51592 |  | 1 (0.0%) | 54 (5.0%) | 55 (5.0%) |
| 10: North Asia | 5151 | 2410 | 19 (0.3%) | 1 (0.0%) | 17 (2.7%) | 37 (2.9%) |
| 11: Central Europe | 3927 | 2092 | 2 (0.0%) |  | 15 (0.5%) | 17 (0.5%) |
| 12: Caucasus and Middle East | 1888 | 1307 |  |  | 2 (0.0%) | 2 (0.0%) |
| 13: Central Asia | 54429 | 49303 | 40 (0.0%) | 117 (0.6%) | 34 (0.6%) | 191 (1.2%) |
| 14: South Asia West | 27988 | 33568 | 46 (0.1%) | 32 (0.0%) | 37 (1.0%) | 115 (1.1%) |
| 15: South Asia East | 13119 | 14734 | 56 (0.3%) | 26 (0.1%) | 12 (0.2%) | 94 (0.6%) |
| 16: Low Latitudes | 2939 | 2341 | 366 (5.7%) | 5 (0.1%) | 12 (0.2%) | 383 (6.0%) |
| 17: Southern Andes | 15908 | 29429 | 178 (0.1%) | 19 (0.4%) | 78 (4.3%) | 275 (4.8%) |
| 18: New Zealand | 3537 | 1162 | 4 (0.0%) | 1 (0.1%) | 11 (0.1%) | 16 (0.2%) |
| TOTAL | 213750 | 613226 | 2972 (2.0%) | 710 (0.1%) | 536 (1.5%) | 4218 (3.6%) |

efficiency). Only 25 glaciers fail due to numerical instabilities. Finally, there is a number of other errors (1.5% of the total area) happening at other stages of the model chain. Examples include errors in processing the geometries or failures in computing certain topographical properties due to invalid DEMs. Altogether, 4218 glaciers (3.6% of the total area) cannot be modelled by OGGM. There are strong regional differences, remote and high latitude regions accounting for most of the errors.

5 **4.3 Volume inversion**

A summary of the volume inversion results is presented in Fig. 10. As expected from theory (Bahr et al., 1997, 2015), our glacier volume estimates approximately follow a power law relationship with the glacier area ($V = cS^\gamma$). The coefficients obtained by a linear fit in log space are close, but not equal to the coefficients computed by Bahr et al. (1997). In particular, the OGGM fit is slightly flatter than the theoretical value (Fig. 10, upper panel), in accordance with empirical coefficients (e.g.





Bahr et al., 2015; Grinsted, 2013). This is an encouraging result, especially because it was reached with the OGGM default settings and without calibration.

The global volume estimates are particularly sensitive to the choice of the ice dynamics parameters, as shown in Fig. 10 (lower panel). As for individual glaciers, the total volume follows an inverse polynomial curve as expected from the equations

of ice flow. Changing from a rectangular to a parabolic bed shape yields a volume loss of exactly 1/3: this is also expected from geometrical considerations[5]. The mixed parabolic/rectangular bed shape model implemented by default therefore lies in between. The three independent estimates plotted as straight dotted lines (VAS; Huss and Farinotti, 2012; Grinsted, 2013) illustrate that $A$ is a relatively straightforward parameter to act upon in order to fit the model to observations. The effect of $A$, however, is going to be the same on all glaciers and therefore will be a poor measure of performance (see also Bahr et al.,

2015, Sect. 8.11). In fact, the added value of OGGM is more likely to be found in the *deviations* from the scaling law (Fig. 10, lower panel). The deviations are the result of a range of possible factors such as slope, total accumulation, or altitude area distribution. With accurate boundary conditions, OGGM should be able to provide more accurate estimates, within the limits of the assumptions and simplifications behind the model equations. The calibration and validation of the OGGM inversion model will be the topic of a subsequent study.

## 4.4 Dynamical runs

We test the model behaviour by running several 300-yr long global simulations under various climate "scenarios". In the first simulations (Fig. 11), we run the model under the climate of the past 31 years. In order to keep the forcing realistic, we create a pseudo-random climate by shuffling the years infinitely. We also run two additional simulations with a 0.5°C positive and negative bias. The unbiased simulation illustrates the committed glacier mass-loss, i.e. the ice mass which is not sustainable

under the current climate. Figure 11 shows that all regions will continue to lose ice even if the climate remains constant. The regions with the most largest committed mass loss relative to the initial volume are Western Canada and US (02), Svalbard (07), and the three High Mountain Asia regions (13, 14, 15). Inversely, the regions Arctic Canada South (04), Greenland (05) and Iceland (06) are least affected. The reasons for these regional differences are complex: they are due to the climate itself of course, but also to glacier properties such as size, slope, continentality. The regions that are far from equilibrium also tend

to be less sensitive to the temperature bias experiments, although this should not be over-interpreted (indeed, the range of the y-axes can hide differences which appear small in comparison to the large regional glacier loss).

In general, the model behaviour looks reasonable and the regional differences are in qualitative agreement with other global studies (e.g. Huss and Hock, 2015, where the regions mentioned above also strike out for their stronger response to 21st century climate change). Also our global estimate of the committed mass loss (approx. 33% at the end of the 300-yr simulation,

probably more at equilibrium) is in agreement with other studies (27±5% and 38±16% for Bahr et al., 2009; Mernild et al., 2013, respectively).

---

[5]we recall that from an ice-flow point of view there is currently no difference between the basal shear stress in a parabolic and a rectangular bed, the two yielding the same ice thickness $h$ but a different volume



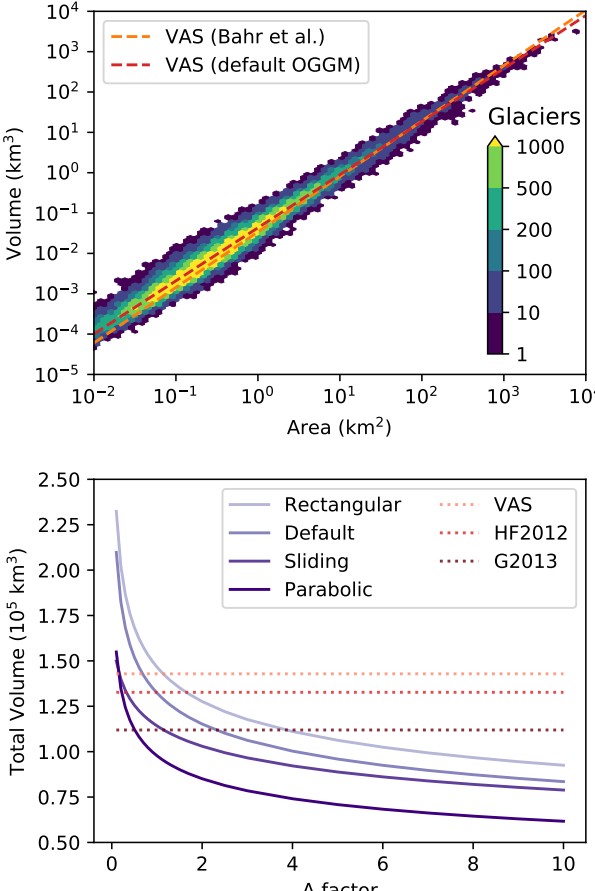

**Figure 10.** Global glacier volume modelling. **Upper panel**: binned scatter plot of volume versus area for all valid glaciers (N=209'532) with the default OGGM setup. Color shading indicates the number of glaciers in each bin. Note the logarithmic scale of the axes and the irregular color scale levels. The dashed lines indicates the volume area scaling relationship with either the theoretical parameters from Bahr et al. (1997) ($V = 0.034\,S^{1.375}$) or fitted on our own data ($V = 0.043\,S^{1.315}$). **Lower panel**: global volume estimates as a function of the multiplication factor applied to the ice creep parameter A, with four different set-ups: defaults, with sliding velocity, with rectangular and with parabolic bed shapes only (instead of the default mixed parabolic/rectangular). In addition, we plotted the estimates from standard volume area scaling ($V = 0.034\,S^{1.375}$), Huss and Farinotti (2012) (HF2012) and Grinsted (2013) (G2013). The two latter estimates are provided for indication only since they are based on a different glacier inventory.





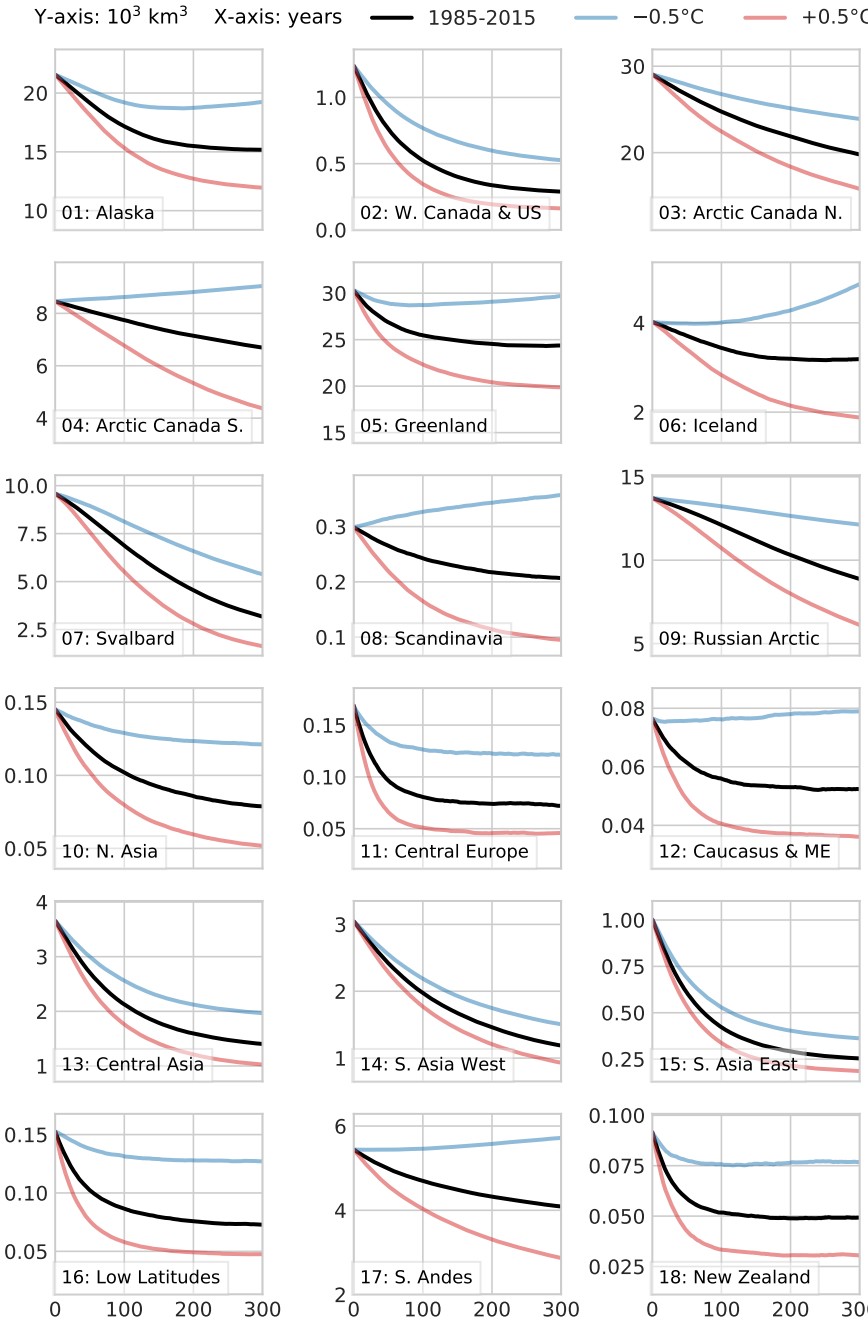

**Figure 11.** Regional glacier volume change under the 1985-2015 climate (randomized) with three temperature biases ($-0.5°$, $0°$, $+0.5°$). Note the units of the y-axes ($10^3$ km$^3$) and the marked regional differences.





A further model test is presented in Fig. 12. Here, we apply a new climate scenario: the climate at t* which, for each glacier individually, represents a theoretical equilibrium climate. In addition to the global response to these scenarios, we separate between the majority group of smaller glaciers and the much smaller group of very large glaciers. Both groups are selected so that they sum up to one quarter of the total glacier volume. A striking feature of the runs is that the glaciers tend to grow under

the artificial t* climate: the growth is slow at first and accelerates with time, hinting towards a positive feedback. This feedback is driven by two factors: first, a higher surface elevation leads to a positive change in mass balance (mass balance-elevation feedback); second, because of the parabolic and trapezoidal bed shapes, a larger ice thickness leads to a wider accumulation area above the ELA and to a wider ablation area below the ELA. It appears that the positive width-accumulation feedback is stronger than the negative width-ablation feedback. This can be explained by the larger accumulation area of glaciers in an

equilibrium climate: the average accumulation area ratio at t* in OGGM is 52%. In order to test which of these feedbacks is stronger, we run a simulation with rectangular bed shapes exclusively (dotted light purple line in Fig. 12), therefore eliminating the width-accumulation but keeping the mass balance-elevation feedback. The results show that for the vast majority of glaciers the feedback disappears entirely, while the very large glaciers still show a weak and delayed altitude feedback.

It is unclear whether this is a bug or a feature. On the one hand, this behaviour is not really desirable since one would expect

glaciers to remain constant under a theoretical equilibrium climate. On the other hand, t* is just a vehicle to calibrate the model and was not supposed to yield a particular insight (for example, many glaciers can only have an equilibrium t* climate after the application of a bias to the operational mass balance model). There are many reasons why small initial perturbations such as numerical noise or the differences between the bed inversion and forward model numerical schemes might lead to a different equilibrium. It must also be noted that this feedback is slow to appear, and will only have a notable influence on the largest

glaciers for long term simulations in a cooler climate (the global volume change after 100 years due to the feedback is 2.7% for the default and 0.4% for the all rectangular cases). The very simple definition of an "equilibrium climate" for these very large glaciers is problematic anyway: large glaciers have a very slow but potentially large response to the smallest changes in climate. At the global scale, most of the 300-yr volume loss is due to the small glaciers, which respond faster and stronger than larger ones.

**5 Conclusions**

We present a new model of global glacier evolution, the Open Global Glacier Model (OGGM, v1.0). The panoply of tools available to compute past and future glacier change range from simple box models (e.g. Harrison, 2013) to more complex, geometry aware models (Huss and Hock, 2015, to cite the most recent in date). OGGM undoubtedly belongs to the complex side of this scale. Different model complexities are justified by different problem settings, taking into account the model-

specific merits and drawbacks. Instead of endorsing one approach over the other, OGGM aims to provide a framework which allows to switch between models and allows objective intercomparisons. In fact, the ice dynamics module represents only a small fraction of the OGGM codebase: a huge amount of work has been invested to provide a series of tools which will help others in their own modelling endeavours. Any interested person can download, install, and run these tools at no cost.





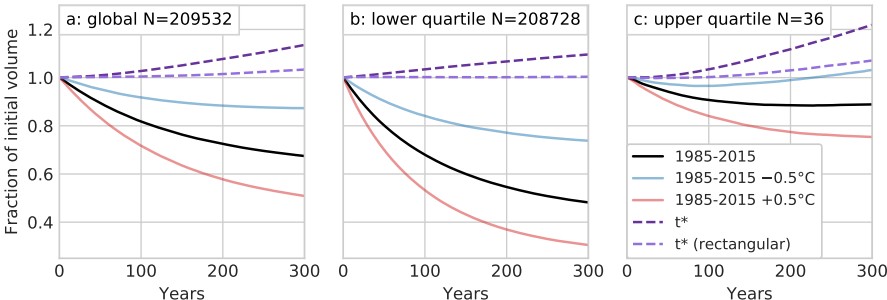

**Figure 12. a**: global glacier volume change under various climate scenarios (1985-2015 climate with three temperature biases and climate at t* which, for each glacier individually, represents a theoretical equilibrium climate) and model configurations (rectangular bed instead of the mixed default), plotted as a fraction of the initial volume. **b** and **c**: volume changes of all glaciers making up for the first and last quartile of the sorted cumulative total volume.

This includes the automated download of topographic and climate data for any location on the globe, the collation of glacier attributes, the automated computation of glacier centerlines, or the delineation of glacier dynamical entities. While some of these tools have been described elsewhere, the added value of OGGM is that they are now centralised, documented, and available for public review via the open-source model.

Accordingly, we cannot demonstrate that OGGM will provide more accurate estimates of future sea-level rise than earlier attempts. However, OGGM allows new studies which weren't possible before. The dynamical representation of glacier advances and retreat enables studies of glacier evolution at long (paleo-) time scales, where ice dynamics and geometrical attributes such as the accumulation area ratio play an important role (e.g. Mackintosh et al., 2017). The modular framework allows to compare the performance of various parametrisations such as the mass balance and downscaling algorithms. It may be argued that the

amount of available data is not enough to constrain modelling studies such as ours at the global scale: OGGM can now be used to test this argument by allowing simpler modules to be added to the codebase and test the added value of increased complexity.

Planned and envisioned future developments for the model follow the general guidelines of modularity and extendability. While some of the authors are working on adding even more complexity to the model (for example by improving the calving and mass balance parametrisations or by implementing a distributed ice dynamics module), it is part of our plans to implement

simpler approaches such as the original Marzeion et al. (2012b) model or the Huss and Farinotti (2012) approach to ice thickness estimation. A considerable amount of work will be needed to correctly assess the uncertainties associated with the model chain: here, Monte Carlo and Bayesian approaches might be the way to follow.

The non-linear dynamical behaviour of glaciers raises a wide range of very interesting inverse problems. For example, how to deal with the transient climate issue in the ice thickness inversion algorithm? How much information about past climate can

be extracted from moraine proxies and today's glacier extent? What are the uncertainties associated with global sea-level rise estimates, and where do they originate? How much complexity is just right? These are all questions the authors hope will be easier to address through the publication of OGGM.





## 6 Code availability, testing, and software requirements

The OGGM software is coded in the Python language and licensed under the GPLV3 free software license. The latest version of the code is available on Github (https://github.com/OGGM/oggm), the documentation is hosted on ReadTheDocs (http://oggm.readthedocs.io), and the project webpage for communication and dissemination can be found at http://oggm.org. Past and
intermediate versions are available in a permanent DOI repository (https://zenodo.org/badge/latestdoi/43965645). The software ships with an extensive test suite which can be used by the users to test their configuration. The tests are triggered automatically at each new code addition, reducing the risk of introducing new bugs (https://travis-ci.org/OGGM/oggm for Linux, https://ci.appveyor.com/project/fmaussion/oggm for Windows). The suite contains unit tests (for example for the numerical core) and integration tests based on sets of real glaciers. At the time of writing, 90% of all relevant lines of code are covered by the
tests (i.e. called at least once by the test suite). The remaining 10% are challenging to monitor because they mostly concern the automated downloading tools which are used in production and cannot be tested automatically.

The following open-source libraries have to be installed in order to run OGGM: `numpy` / `scipy` (Van Der Walt et al., 2011), `scikit-image` (van der Walt et al., 2014), `shapely` (Gillies, 2007), `rasterio` (Gillies, 2013), `pandas` (McKinney, 2010), `geopandas`, `xarray` (Hoyer and Hamman, 2017), `pyproj`, `matplotlib` (Hunter, 2007), and `salem` (Maussion
et al., 2017). OGGM runs on all major platforms (Windows, Mac, Linux) but we recommend to use Linux as this is the platform it is most tested on.

## Appendix A: Climate data

The default climate dataset used by OGGM is the Climatic Research Unit (CRU) TS v4.01 Dataset (Harris et al., 2014, released 20.09.2017). It is a gridded dataset at 0.5° resolution covering the period 1901-2016. The dataset is obtained by interpolating
station measurements and therefore does not cover the oceans and Antarctica. The TS dataset is further downscaled to the resolution of 10' by applying the 1961-1990 anomalies to the CRU CL v2.0 gridded climatology (New et al., 2002). This step is necessary because the TS datasets do not contain an altitude information, which is needed to compute the temperature at a given height on the glacier. To compute the annual mass balances we use the hydrological year convention (the year 2001 being October 2000 to September 2001 in the Northern Hemisphere, April 2000 to March 2001 in the Southern Hemisphere).
For each glacier, the monthly temperature and precipitation time series are extracted from the nearest CRU CL v2.0 grid point and then converted to the local temperature according to a temperature gradient (default: 6.5K km$^{-1}$). No vertical gradient is applied to precipitation, but we apply a correction factor $p_f$=2.5 to the original CRU time series (similar to Marzeion et al., 2012b). This correction factor can be seen as a global correction for orographic precipitation, avalanches, and wind-blown snow. It must be noted that this factor has few (if any) impact on the mass balance model performance in terms of *bias*. This
is due to the automated calibration algorithm, which will adapt to a new factor by acting on the temperature sensitivity $\mu^*$. To verify that the chosen precipitation factor is realistic, we use another metric: the standard deviation of the mass balance time-series. Comparisons between model and observations show that the model underestimates variability by about 10%: we





could tune the precipitation factor towards higher values to reduce this discrepancy but refrain to do so, as we do not want to add an additional free parameter in the model.

## Appendix B:  WGMS glaciers

To calibrate and validate the mass balance model, OGGM relies on mass-balance observations provided by the World Glacier
Monitoring Service (WGMS, 2017). The Fluctuations of Glaciers (FoG) database contains annual mass-balance values for several hundreds of glaciers worldwide. We exclude tidewater glaciers and the time series with less than five years of data. Not all of the remaining glaciers can be used by OGGM: we also need a corresponding RGI outline. Indeed, the WGMS and RGI databases have distinct glacier identifiers and it is not guaranteed that the glacier outline provided by the RGI fits the outline used by the local data providers to compute the specific mass balance. Since 2017, the WGMS provides a lookup table linking
the two databases. We updated this list for the version 6 of the RGI, leaving us with 254 mass balance time series.

These data are not equally distributed over the glaciated regions (see e.g. Zemp et al., 2015), and their quality is highly variable. In the absence of a better data basis (at least for the 20th century), we have to rely on them for the calibration and validation of our model. Fortunately, these data play only a minor role in the model calibration as explained in Sect. 3.3. For future studies it might be advisable to use independent, regional geodetic mass balance estimates for validation as well.

## 15   Appendix C:  RGI Regions

A map of the RGI regions and some basic statistics are presented in Fig. C1.

*Author contributions.*  BM and FM are the initiators of the OGGM project. FM is the main OGGM developer and wrote most of the paper. AB developed the downstream bedshape estimation algorithm. JE developed the glacier partitioning tool. KF wrote parts of the bed inversion and dynamical cores. AJ provided a robust implementation of the dynamical core used for testing and contributed to the development of the
operational scheme. JL provided the WGMS to RGI lookup table and contributed to the topographical data download tool. FO contributed to the AWS deployment tool. BR developed the calving parametrisation tool. TR developed the download and parallelisation tools and is largely responsible for the successful deployment of OGGM on supercomputing environments. AV contributed to the climate and mass balance tools. CW provided the first implementation of the centerline determination algorithm. All authors continuously discussed the model development and the results together.

*Acknowledgements.*  BM, JL, and CW were supported by the Austrian Science Fund (FWF), grant P25362. BM, AB, and JE were supported by the German Research Foundation, grant MA 6966/1-1. AV and BR were supported by the DFG through the International Research Training Group IRTG 1904 ArcTrain. The computations were realised partly on resources provided by Amazon Web Services Cloud Computing (sponsored by Amazon) and on the computing facilities of the Institute of Geography, University of Bremen.





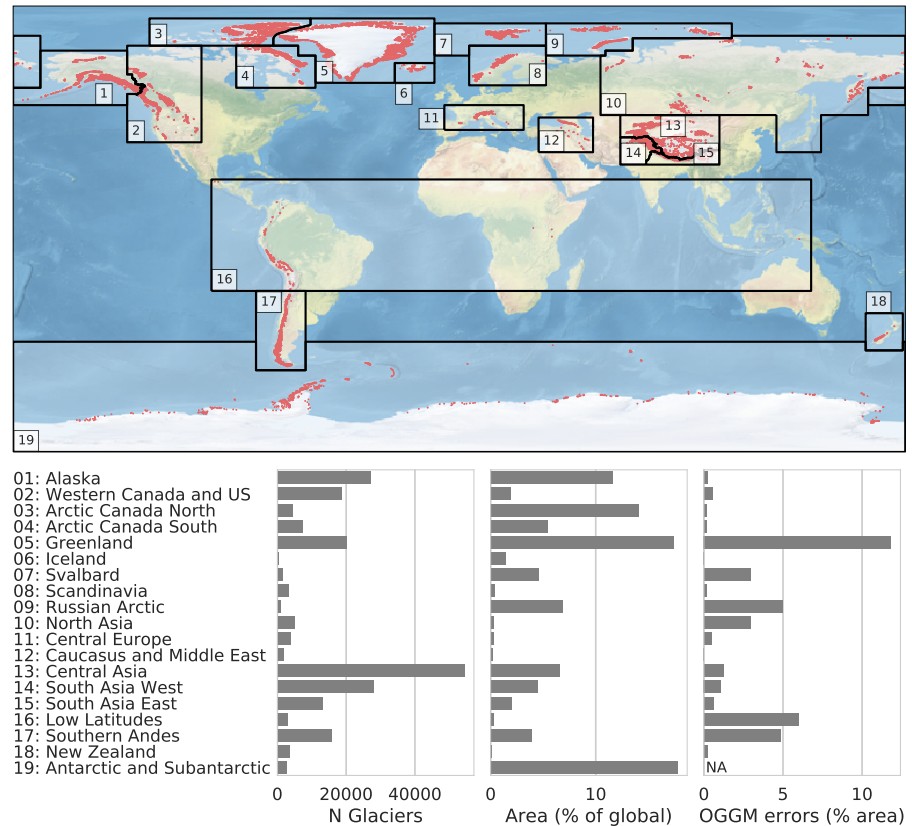

**Figure C1. Upper panel**: map of the RGI regions; the red dots indicate the glacier locations. **Lower panel**: region names and basic statistics of the database (number of glaciers per region, regional contribution to the global area in percent, and percentage of the regional area which cannot be modelled by OGGM).

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
