# Peer review of "The Open Global Glacier Model (OGGM) v1.1"

_Geoscientific Model Development, 2018_

## Referee Comment (RC1) · D. Rounce (Referee) · 18 Apr 2018

**Review of "The Open Global Glacier Model (OGGM) v1.0"**
**by Maussion et al.**

General Comments

The study describes a new open source global glacier evolution model named Open Global Glacier Model (OGGM) that is written in Python. The paper is very well written and provides a good balance of model details and examples in addition to descriptions of model limitations and areas of future work. One of the best attributes of the model is its modular framework, which should help facilitate a comparison of how different aspects of the model impact the results, and as the authors state, will help answer many questions about global glacier modeling like what amount of modeling complexity is just right. The model also fills a major gap in the global mountain glacier modeling community as an open source community-driven model currently does not exist like those in the ice sheet community. In my opinion, the pre-processing tools alone will be a very valuable contribution to the mountain glacier community and the modular framework should enable the model to be community-driven. For these reasons, I recommend accepting this paper with minor revisions.

These minor revisions mostly deal with grammar and making a few sections a bit easier to understand. Additionally, the authors use the term "calving" throughout the manuscript, which might be better replaced by the term "frontal ablation", which includes calving and other mass loss processes that occur for marine/lake-terminating glaciers. Lastly, the authors state that the model is intended to be community-driven and identify many places in the manuscript where future work/modules will be developed; however, there does not appear to be much discussion of how users in the community outside of the model development team could contribute to future model development. This does not have to be discussed in detail, but I would encourage the authors to add a few lines to Section 6 about how they envision users who are not affiliated with the OGGM team to be able to contribute new modules. Specific comments are detailed below.

Specific Comments
P refers to page
L refers to line number
*Italics* indicate suggested grammatical changes

P1 L2 – Delete "of" to read "Despite their importance…"

P1 L2 – "… and *being a* source of geohazards…"

P1 L12 – Delete "a" to read "the model shows very realistic behavior"

P1 L16 – "… added to the codebase, *which allow* new kinds of model intercomparisons *to be run* in a controlled environment …"

P1 L18 – The future developments describe new physical processes and model calibration, but what about other community-driven efforts that perhaps have not been thought of yet? Perhaps a phrase could be added acknowledging these other unforeseen developments.

P2 L12 – "… improve the knowledge *of* how glaciers …"

P2 L20 – comma after unfortunately can be deleted

P3 L21 – Use of a comma instead of apostrophe for number of inventoried glaciers?

P3 L26 – I would recommend using "frontal ablation" instead of "calving" (see general comment).

P4 L2 – Perhaps future improvements and new modules?  See general comment about community-driven model, but I think there could be value in adding various modules that model mass balance or glacier dynamics in different ways.

P4 L4 – Why "the" Tasman Glacier?  Is it always referred to in literature as "the Tasman glacier"?  If not, then the can be deleted.

P4 L23 – In the current form it appears that the mass balance model and the glacier evolution are completely separate.  Is that the case or does the model compute the mass balance for a given timestep (month, year, etc.) and then allow the glacier evolution to occur?  If it is the latter, then I would suggest stating how the mass balance and glacier evolution are linked in the model.

P5 Figure 1 – Difficult to see the difference between c and d.  Do c and d have a scale bar?  Is this the scale bar associated with e and f?  Or are the color of all the transects associated with the scale bar next to f?  I assume the latter is the case, but perhaps explicitly stating this in the caption could be helpful.

P6 L26 – What do you mean by "because of the lack of traceability"?  Do you mean that the dates when the DEMs were acquired or the data they were generated from are unknown?

P7 L3 – Is there a reason for the spatial resolution of the target grid varying for each glacier, but then the Gaussian filter being applied at a constant 250 m radius?  If the grid size is varied, then why is the filter size not varied as well?

P7 L8 – "… because it allows *one* to compute centerlines and define…"

P7 L11 – "… minor role *compared* to …"

P7 L14 – I'm a bit confused by the default grid spacing.  Is this still a 2d "grid" or is this now referring to a "line" spacing of the transects.  Also, does map topography refer to the resolution of the DEM?  Is this the same spatial resolution of the target grid (P7 L1)?

P7 L20 – Why is this necessary?  Is this done to avoid sinks, to ensure that each flowline only has a single flux upstream contributing to it (although at glacier branches there would be multiple anyways), or is this done to reduce problems associated with the ice thickness inversion?  Please clarify.

P7 L26-32 – Are these the same steps as those associated with Figure 1? I would like to note that each step is much easier to visualize here in Figure 2.

P7 L31 – "… of the glacier is *the* exact same…"

P9 L5 – the global precipitation correction factor is a constant of 2.5 according to Appendix A. Since it does not vary regionally, I would recommend stating its value here as well.

P9 L9 – Does $\mu^*$ have a name?

P9 L15 – "… justifies describing it here."

P10 L7 – How far away can this be? Is there a limit for how close these 10 locations need to be? Could you add the largest distances for example?

P10 L8 – reference Figure 3 when describing Hintereisfener

P10 L11 – Since the residual bias is subtracted from the modelled mass balance, should this residual bias be added to equation 1? Any information on what percentage of glaciers this is required for? If it's a small percentage, then perhaps it's not important to include in equation 1.

P12 L10 – change comma to decimal point

P13 L11 – "The equation varies *by* a …"

P13 L22 – Can this overestimation be quantified or is it so reliant on the creep parameter that it doesn't make sense to add a range of values here? If it can be quantified (perhaps only for the default value), then it would be nice information to include.

P13 L32 – "is consequent" doesn't really make sense. I would suggest not negligible, sizable, noticeable, impactful, or something similar instead.

P14 Figure 6 – The order of (a) and (b) in the captions comes after they are described, while in Figure 5 they come before the subfigure is described. I would change these to be before in order to be consistent throughout the figure captions.

P15 L15 – What does "translate in a certain h" mean? I assume it means that a new h is calculated? Furthermore, "allowing to have" also sounds awkward. Perhaps "which allows" or "which enables" various bed geometries to be present along the same flowline?

P15 L28 – Consider using advancing instead of growing, since in this case the changes to the downstream flowline enables the glacier to advance as opposed to simply growing, which could refer to a glacier that is only increasing its ice thickness and not advancing.

P17 L4 – "*They* are considered…" and "e.g." can be deleted.

P17 L14 – "… a valley glacier is *the additional* mass loss *that occurs* at the ..." See general comment about frontal ablation instead of calving as well.

P18 L3 – "… has the advantage *of preventing* tidewater glaciers *from advancing* while … they stop *calving*)."

P18 L20-22 – These sentences refer to the RGI polygons, but these polygons are partially covered in Figure 7. Is it possible to make the polygons clearer?

P20 L6 – Perhaps "section" instead of "chapter"

P22 L21 – "most largest" doesn't make sense. I recommend deleting most.

P22 L24 – "… size slope, *and* continentality"

P22 L28 – What is meant by "strike out"? Please clarify.

P22 footnote – Perhaps use "we note" instead of recall

P23 Figure 10 – specify that VAS is referring to volume area scaling as the others have been described.

P25 L31 – "allows *one* to switch"

P25 L33 – Here is another opportunity where perhaps you could state that the interested person could also contribute to the development of new modules as well to reinforce that this is meant to be a community-driven model? See general comment.

P26 L8 – "allows *one* to compare"

P27 L15 – "we recommend *using* Linux"

---

## Short Comment (SC1) · 10 May 2018

We would like to thank Reviewer 1 (David Rounce) for his thoughtful and constructive comments about our manuscript. A detailed point by point response will follow once the other review(s) are available, but we would like to clarify two valid (and important) points raised in the review.

**1  External contributions**

You write: "*Lastly, the authors state that the model is intended to be community-driven and identify many places in the manuscript where future work/modules will be developed; however, there does not appear to be much discussion of how users in the*

[Figure]

*community outside of the model development team could contribute to future model development.*"

This is an important point and we will include more discussion in the revised manuscript. As model providers/developers, we can only encourage the community to contribute. We attempt to do so by several means:

1. it must be relatively easy for a new user to detect where and how his/her contribution can be implemented

2. the model must be able to cope with different ways to represent/simulate the considered process

3. we must ensure attribution to the original contribution (e.g. a scientific publication)

**For point 1**, documentation and code clarity is key. We have done our best to make the model accessible and understandable via the online documentation, but we are aware that there are still some rough edges. Furthermore, a good knowledge of the Python language is necessary before being able to contribute. In this respect, OGGM isn't very different from other models written in FORTRAN or C, but we plan to provide as much support as possible to the future contributors of the model.

**For point 2**, we think that the current structure of the model allows a relatively efficient modularity. Since every task in the workflow writes and reads the data from disk, tasks can be replaced/enhanced at which, as long as the format of the input/output files is agreed beforehand. The modularity will never be perfect, of course, and we expect that the model internals will have to be adapted in order to accept new contributions when they come.

**Point 3** is something we didn't consider until recently. Attribution is important in the scientific community, for many reasons. Therefore, we now make the following suggestion to the interested contributors:

- if the changes are small or concerning the model internal structure, they should be proposed to the main codebase

- if the changes concern an entire part of the model workflow (e.g. a new ice thickness inversion model, or a new mass-balance model), then they can be either added to the central codebase or maintained in an external repository. The latter solution has the advantage that it guaranties freedom of development and a correct attribution to the original contributor of the module.

In fact, your comment motivated the development of a template repository for external OGGM modules. Interested users will find this repository on GitHub: https://github.com/OGGM/oggmcontrib and the documentation on ReadTheDocs: http://oggmcontrib.readthedocs.io/en/latest/

We hope that this will foster new collaborations!

**2 Coupling of the mass-balance and dynamical models**

You write: "*In the current form it appears that the mass balance model and the glacier evolution are completely separate. Is that the case or does the model compute the mass balance for a given timestep (month, year, etc.) and then allow the glacier evolution to occur?*"

You are right, we did not specify this point in the manuscript. The coupling between the two models is a user choice. The mass-balance profile used by the dynamical model can be updated:

- at each time-step of the dynamical model (e.g. daily timescales)

- each month

- each mass-balance year (the default)

- only once (for testing / sensitivity analysis purposes)

In practice, this doesn't make much difference at the time scales relevant for ice dynamics (decades to centuries), and the choice of a yearly update is mostly driven by performance considerations. The model is tested with all three options though, and the results are indeed very close.

Note that this this doesn't mean that the mass-balance model cannot compute the mass-balance at shorter time intervals if required by the physical parameterizations. The interface between the model elements simply requires the mass-balance model to integrate the mass-balance over a year before giving it to the dynamical model.

We will clarify this point in the manuscript.

---

## Referee Comment (RC2) · Anonymous Referee #2 · 7 Sep 2018

I am sorry for this late review - the summer was distracted and this manuscript was not one to take lightly. Indeed, this paper represents a huge and ambitious undertaking, one that represents a huge leap forward in modelling global glacier response to climate change. I have nothing but admiration for the authors and their contribution. I suspect this paper will become one of the most cited contributions in GMD, as this advance in glacier modelling makes most efforts to date obsolete. As someone that has worked hard on similar kinds of modelling of individual glaciers, the effort to create a state-of-the-art, open-source model that can be applied to 10s of 1000s of glaciers on a global scale is remarkable.

While there are many considerations for model improvement - one could quibble with different simplifications and assumptions - the authors point out most of these themselves, and one needs to start somewhere. This is already an extremely sophisticated treatment on many fronts. I will make only minor comments here. Perhaps only point 5

is significant, if I am understanding correctly.

1. The English is often incorrect or awkward, starting with the opening of the abstract: this should be "Despite their importance...." I will not make a comprehensive review of the grammatical suggestions, but a few other points are noted below.

2. p.2, l.33, reference to Elmer/Ice in the ice sheet modelling community. This has also been widely applied to mountain glacier modelling - this should be noted. Also, there are other public-domain ice sheet models such as glimmer (in the CISM family) and ISSM.

3. p.4, l.8, this might be a suitable juncture to note SRTM's resolution ($\sim$90 m) and the year that these elevations date to (is it 2000?). On this topic, is this reasonably well-matched with the RGI outlines? I assume the latter span a range of years - worth summarizing here.

4. p.6, I see that DEM resolution are summarized here, so that maybe covers off the first part of item 3. It would be worth reporting the years for each DEM though - what year(s) does the reference glacier hysometry then refer to?

5. p.7, ll.1-2. I am confused by the square relation, $dx = aS^2$. Maybe I misunderstand, but should this not be a square-root relation? $dx = aS^{(1/2)}$. Dimensionally and conceptually.

6. p.11, l.24, "modelling glacier evolution" - you mean thickness, I think

7. p.18, discussion of ice caps. Does this also refer to alpine icefields, I assume? Mountain complexes with a shared accumulation area and multiple outlets. Please clarify.

8. p.22, l.10 and the discussion around this. The deviations from the scaling law seem consistent with the results of Adhikari and Marshall (GRL, 2012), which was dismissed by Bahr et al (2015). Is it fair to say that the results here are consistent with the expectation that a variety of local factors such as sliding, glacier cross-sectional

shape, mass balance profile, and state of disequilibrium can cause a different scaling relationship, vs. a kind of universal constant for the scaling-law exponent as argued by Bahr et al.?

A few minor points

p.1, l.14, "several dozen glaciers"

p.2, l.3, "superimposed" does not seem quite right. "masked"?

p.3, l.21, "every single one of" or "all of"

p.13, l.15, "these", not "theses"

---

## Author Comment (AC1) · 14 Nov 2018

We are very thankful to both reviewers for their thoughtful comments on our manuscript. We also would like to apologize for the unusually long time we needed to provide this answer: several model developments happened since our first submission, and we wanted to make sure that they are operational before revision. We are now confident that we can provide a revised manuscript together with a new model version (OGGM v1.1) within the next few weeks.

These changes are described below:

- The method used to apply gridded anomalies to the climatology has changed for precipitation: we now use scaled anomalies (correction factor) instead of the standard anomalies (which in very rare cases could lead to negative precipitation

anomalies). We will add a sentence in "Appendix A: Climate data" to reflect this change.

- The temperature sensitivity $\mu^*$ is now attributed to each flowline (instead of one glacier-wide $\mu^*$ previously). This change allows for more flexibility in the calibration procedure, and allows to merge neighbouring glaciers as they grow. This change is not relevant for the manuscript or for future simulations, but it is very useful for past (paleo-) simulations where glaciers were much larger than today.

- The ice dynamics and bed inversion modules now have an option to account for lateral drag depending on the shape of the flowline's bed. The parameterization is based on Adhikari & Marshall (2012) and it is turned off per default. We will add a discussion about the sensitivity of the model to this new parametrization in Figs. 6, 8 and 10. The implementation was done by Philipp Gregor, and he will be a new co-author of the revised manuscript.

- We have worked extensively towards increasing the transparency of the model's results and reproducibility. This was done by continuing our work on model documentation, and the creation of dedicated tools for continuous monitoring of the model results. One of these tools is a benchmark tracking both the model execution time and model results (https://cluster.klima.uni-bremen.de/~github/asv/#/), with the goal to monitor and detect undesired changes in model behaviour. The other is a continuous monitoring and visualization of the mass-balance model comparison to individual glacier mass-balance (beta-version: https://cluster.klima.uni-bremen.de/~mdusch/ci/). The implementation was done by Matthias Dusch, and he is now a co-author of the revised manuscript.

- A first publication based on OGGM originating from a working group outside of the "core team" is now available: Goosse et al. (2018).

- And other changes too numerous to be listed here. For the full list of model

improvements, see https://docs.oggm.org/en/latest/whats-new.html

These changes were not all requested by the reviewers but were the result of a standard model development happening as the model gains more users. They will not change our manuscript or the model results in a significant way.

We now provide a point by point answer to the reviewers comments. ("RC" stands for "reviewer comment", "AR" for "authors response")

**Referee 1 (David Rounce)**

**Community driven development**

You write:

*"RC: the authors state that the model is intended to be community-driven and identify many places in the manuscript where future work/modules will be developed; however, there does not appear to be much discussion of how users in the community outside of the model development team could contribute to future model development. This does not have to be discussed in detail, but I would encourage the authors to add a few lines to Section 6 about how they envision users who are not affiliated with the OGGM team to be able to contribute new modules."*

*"RC: P1 L18 - The future developments describe new physical processes and model calibration, but what about other community driven efforts that perhaps have not been thought of yet? Perhaps a phrase could be added acknowledging these other unforeseen developments."*

*"RC: P4 L2 – Perhaps future improvements and new modules? See general comment about community-driven model, but I think there could be value in adding various*

*modules that model mass balance or glacier dynamics in different ways."*

*"P25 L33 – Here is another opportunity where perhaps you could state that the interested person could also contribute to the development of new modules as well to reinforce that this is meant to be a community-driven model? See general comment."*

**AR:** see our previous response (10 May 2018). We will add a paragraph in the discussion.

**Other comments**

We addressed all editorial comments (thanks!) and do not list them here. The remaining open questions are:

*"RC: I would recommend using frontal ablation instead of calving (see general comment)."*

**AR:** Agreed. We replaced all occurrences in the text.

*"RC: P4 L23 – In the current form it appears that the mass balance model and the glacier evolution are completely separate. Is that the case or does the model compute the mass balance for a given timestep (month, year, etc.) and then allow the glacier evolution to occur? If it is the latter, then I would suggest stating how the mass balance and glacier evolution are linked in the model."*

**AR:** You are right, we did not specify this point in the manuscript. The coupling between the two models is a user choice. The mass-balance profile used by the dynamical model can be updated: (i) at each time-step of the dynamical model's computation (e.g. daily timescales, depending on the chosen time step), (ii) each month, (iii) each mass-balance year (the default), or (iv) only once (for testing and mass-balance feedback sensitivity analysis purposes)

In practice, this doesn't make much difference, and the choice of a yearly update is

mostly driven by performance considerations. The model is tested with all three op-
tions.

Note that this doesn't mean that the mass-balance model cannot compute the mass-
balance at shorter time intervals if required by the physical parameterizations. The
interface between the model elements simply requires the mass-balance model to in-
tegrate the mass-balance over a year before giving it to the dynamical model.

We will clarify this point in the manuscript.

*"RC: P5 Figure 1 – Difficult to see the difference between c and d. Do c and d have
a scale bar? Is this the scale bar associated with e and f? Or are the color of all the
transects associated with the scale bar next to f? I assume the latter is the case, but
perhaps explicitly stating this in the caption could be helpful."*

**AR:** Indeed, this figure can be hard to interpret. Figs. 1c and 1d do not have a scale
bar: the colors simply serve the purpose of differentiating them. The difference be-
tween the two is subtle: in 1c, the lines are the "geometrical" ones, i.e. computed from
the intersection of the flowline's normal line with the outline. In 1d, the lines widths
are corrected to accurately represent the true glacier area-elevation distribution. Their
representation is now centred on the flowline, which is only visible if one looks carefully
(the differences between Fig 2b and 2c are easier to see, but explained later in the
text). We will add an explanation to the caption and refer to the later explanation in the
text.

*"RC: P6 L26 – What do you mean by "because of the lack of traceability"? Do you
mean that the dates when the DEMs were acquired or the data they were generated
from are unknown?"*

**AR:** yes. It can probably be tracked down, but not easily – this information is scat-
tered around the website and probably incomplete. Visually, the DEMs look good and
realistic, but biases are possible. We are aware that this is a suboptimal solution, but

this really isn't in our control as long as no consistent, gap-free global DEM is available. Unfortunately, the newly released Tandem-X DEM (https://geoservice.dlr.de/web/dataguide/tdm90/) also seems to have many data gaps. We will continue to explore better solutions in the future.

*"RC: P7 L3 – Is there a reason for the spatial resolution of the target grid varying for each glacier, but then the Gaussian filter being applied at a constant 250 m radius? If the grid size is varied, then why is the filter size not varied as well?"*

**AR:** the DEM nominal resolution is approx. 90m (the local glacier grid is then interpolated from it). The smoothing is meant to be applied to the nominal DEM, but it is technically easier to do on the interpolated DEM. To be honest, we don't really know the influence of the smoothing radius on the results, and it might play a non-negligible role at the individual glacier scale. This is a tunable model parameter though, and users can try and test it. The change you suggest could be implemented easily, but testing its influence on the model results is the time consuming part...

*"RC: P7 L14 – I'm a bit confused by the default grid spacing. Is this still a 2d "grid" or is this now referring to a "line" spacing of the transects. Also, does map topography refer to the resolution of the DEM? Is this the same spatial resolution of the target grid (P7 L1)?"*

**AR:** Yes, the default grid spacing of "twice that of the map topography" refers to the flowline grid. The "map topography" refers to the local grid (i.e. of varying spacing depending on glacier size). As an example: Hintereisferner has a local map grid spacing of 50m, so its flowlines have a grid spacing of 100m. We will check the semantics throughout the manuscript and will consistently use "line grid", "local map grid" and "nominal DEM grid" in order to avoid such confusion.

*"RC: P7 L20 – Why is this necessary? Is this done to avoid sinks, to ensure that each flowline only has a single flux upstream contributing to it (although at glacier branches there would be multiple anyways), or is this done to reduce problems associated with*

[Figure]

*the ice thickness inversion? Please clarify."*

**AR:** yes: the purpose of this correction is to avoid sinks ("deepenings" in the manuscript, we will now use "sinks" instead). Sinks along a flowline are unphysical: the inversion procedure can cope with them by artificially forcing a positive slope, but this leads to results which are incompatible with the forward dynamical model, which will compensate by filling the sinks with ice before anything else. Where this happens, this would lead to undesirable spin-up issues. We will explain this more clearly in the manuscript.

*"RC: P7 L26-32 – Are these the same steps as those associated with Figure 1? I would like to note that each step is much easier to visualize here in Figure 2."*

**AR**: yes, indeed. We will add a reference to Figure 2 in the caption of Figure 1.

*"RC: P9 L5 – the global precipitation correction factor is a constant of 2.5 according to Appendix A. Since it does not vary regionally, I would recommend stating its value here as well."*

**AR:** agreed.

*"RC: Does $\mu^*$ have a name?"*

**AR:** we use "temperature sensitivity" throughout the manuscript. We are not aware of any general naming convention for this parameter. In the model code and in the OGGM community, we like to call it "mu star".

*"RC: P10 L7 – How far away can this be? Is there a limit for how close these 10 locations need to be? Could you add the largest distances for example?"*

**AR:** no, there is no limit. The inverse distance algorithm ensures that if one glacier is very close and all the others are very far, only the close glacier will count in the weighting. However, a visual estimation based on the map in https://oggm.org/2017/02/19/wgms-rgi-links/ leads to several thousands of kilometres distance for the Russian

Arctic. This alone is an argument for the use of the OGGM-specific $t^*$, instead of the interpolation of any other "physical parameter" such as $\mu$ or similar.

We will add this information in the manuscript and will add the location of the WGMS reference glaciers on the global map in Appendix C for reference.

*"RC: P10 L11 – Since the residual bias is subtracted from the modelled mass balance, should this residual bias be added to equation 1? Any information on what percentage of glaciers this is required for? If it's a small percentage, then perhaps it's not important to include in equation 1."*

**AR:** you are right, this should be included in Eq. 1. And no, it is not negligible at all - locally, it can rise to $\pm$ 1 m w.e. Currently, it is not a tuning parameter (it is a residual of the calibration), but we hope to be able to use regional geodetic estimates to better constrain this value in the future.

*"RC: P13 L22 – Can this overestimation be quantified or is it so reliant on the creep parameter that it doesn't make sense to add a range of values here? If it can be quantified (perhaps only for the default value), then it would be nice information to include."*

**AR:** for this particular idealized case (a glacier of 5km length in strong disequilibrium with its climate) the error is of about 25%. Assuming that the MB gradient and the creep parameter are known, this overestimation will depend mostly on how "strong" this disequilibrium really is. Estimating this number for real case glaciers could be an interesting study, although this effect will be very difficult to disentangle from other uncertainties.

We will add a sentence in the manuscript explaining this number.

*"RC: P15 L15 – What does "translate in a certain h" mean? I assume it means that a new h is calculated? Furthermore, "allowing to have" also sounds awkward. Perhaps "which allows" or "which enables" various bed geometries to be present along the same*

*flowline?"*

**AR:** yes. We will follow your suggestion.

*"RC: P18 L20-22 – These sentences refer to the RGI polygons, but these polygons are partially covered in Figure 7. Is it possible to make the polygons clearer?"*

**AR:** yes, we will make them more visible.

*"RC: P22 L28 – What is meant by "strike out"? Please clarify."*

**AR:** sorry, we meant that the regions with a stronger response to 21st climate change are the same in our study and that of Huss & Hock. We will clarify.

**Referee 2**

**General comment**

*"RC: this paper represents a huge and ambitious undertaking, one that represents a huge leap forward in modelling global glacier response to climate change. I have nothing but admiration for the authors and their contribution. I suspect this paper will become one of the most cited contributions in GMD, as this advance in glacier modelling makes most efforts to date obsolete. As someone that has worked hard on similar kinds of modelling of individual glaciers, the effort to create a state-of- the-art, open-source model that can be applied to 10s of 1000s of glaciers on a global scale is remarkable."*

**AR:** thank you very much for your assessment and support. We would like to add that we don't think that OGGM will make other efforts "obsolete". We sincerely hope to be able to engage a wider community under the OGGM framework, making of OGGM an "umbrella" more than a model that competes with others. We are happy to see that

this idea is gaining momentum, at least in the wider climate community: after a long period without funding, we are now happy to have several people working on the use and development of the model, and external groups have been successfully applying for research funds based on OGGM (University of Louvain).

**Point by point answer**

*"RC: 1. The English is often incorrect or awkward, starting with the opening of the abstract: this should be "Despite their importance...." I will not make a comprehensive review of the grammatical suggestions, but a few other points are noted below."*

**AR:** thanks for your suggestions. Referee 1 also made a number of edits, and we hope to be able to get help from the Copernicus editing team (or a native speaker colleague) before final publication.

*"RC: p.2, l.33, reference to Elmer/Ice in the ice sheet modelling community. This has also been widely applied to mountain glacier modelling - this should be noted. Also, there are other public-domain ice sheet models such as glimmer (in the CISM family) and ISSM."*

**AR:** thanks, we will add these references.

*"RC: 3. p.4, l.8, this might be a suitable juncture to note SRTM's resolution ( âĹij90 m) and the year that these elevations date to (is it 2000?). On this topic, is this reasonably well-matched with the RGI outlines? I assume the latter span a range of years - worth summarizing here."*

**AR:** yes, the outlines and topography do not always match. RGI outlines span several decades (with a large majority of outlines valid for the period 2000-2010) and the SRTM acquisition was in 2000. Since these dates change with each glacier and each DEM it is hard too summarize them all, but they are stored as attributes for each glacier during an OGGM run.
*"RC: 4. p.6, I see that DEM resolution are summarized here, so that maybe covers off the first part of item 3. It would be worth reporting the years for each DEM though - what year(s) does the reference glacier hysometry then refer to?"*

**AR:** yes, we will add the dates of each DEM dataset (when available: see our comment about DEM3 in our answer to Referee 1).

*"RC: 5. p.7, ll.1-2. I am confused by the square relation, $dx = aS^2$. Maybe I misunderstand, but should this not be a square-root relation? $dx = aS^{\frac{1}{2}}$. Dimensionally and conceptually."*

**AR:** you are right: fortunately, the error was in the manuscript, not in the code. We will correct the equation accordingly.

*"RC: 7. p.18, discussion of ice caps. Does this also refer to alpine icefields, I assume? Mountain complexes with a shared accumulation area and multiple outlets. Please clarify."*

**AR:** yes, this applies to any glacier that shares an ice divide with others (e.g. "glacier complexes" in Kienholz et al. 2013). However, we think that most glacier complexes are less problematic than ice-caps, mostly because the ice divides are more obvious in complex topography than flat ice. We will clarify this point in the manuscript.

*"RC: 8. p.22, l.10 and the discussion around this. The deviations from the scaling law seem consistent with the results of Adhikari and Marshall (GRL, 2012), which was dismissed by Bahr et al (2015). Is it fair to say that the results here are consistent with the expectation that a variety of local factors such as sliding, glacier cross-sectional shape, mass balance profile, and state of disequilibrium can cause a different scaling relationship, vs. a kind of universal constant for the scaling-law exponent as argued by Bahr et al.?"*

**AR:** thanks a lot for this comment. Due to the model description nature of our manuscript, we would not like to go this path and will not make such an analysis or

statement in the paper. That said, there is a lot that can be done with our global sensitivity analyses, and yes indeed the scaling law parameters change (sometimes in surprising ways) with the chosen model parametrisation. If you are interested in a follow-up study on this topic, do not hesitate to reach out to us for further discussion.

References

Adhikari, S. and Marshall, S., 2012, Parameterization of lateral drag in flowline models of glacier dynamics, J. Glaciol., 58, 212, 1119–1132, doi: 10.3189/2012JoG12J018

Goosse, H., Barriat, P.-Y., Dalaiden, Q., Klein, F., Marzeion, B., Maussion, F., Pelucchi, P., and Vlug, A.: Testing the consistency between changes in simulated climate and Alpine glacier length over the past millennium, Clim. Past, 14, 1119-1133, https://doi.org/10.5194/cp-14-1119-2018, 2018.

Kienholz, C., Hock, R., and Arendt, A. a. (2013). A new semi-automatic approach for dividing glacier complexes into individual glaciers. Journal of Glaciology, 59(217), 925–937. https://doi.org/10.3189/2013JoG12J138

---

## Author Response (AR1)

**Point by point response to the reviewers - revision**

We are very thankful to both reviewers for their thoughtful comments on our manuscript. We also would like to apologize for the unusually long time we needed to provide this revised manuscript: several model developments happened since our first submission, and we wanted to make sure that they are operational before revision. In our revised manuscript we now present a new model version (OGGM v1.1) with many improvements:

- The method used to apply gridded anomalies to the climatology has changed for precipitation: we now use scaled anomalies (correction factor) instead of the standard anomalies (which in very rare cases could lead to negative precipitation anomalies). At the global scale this had almost no effect.

- The temperature sensitivity $\mu^*$ is now attributed to each flowline (instead of one glacier-wide $\mu^*$ previously). This change allows for more flexibility in the calibration procedure, and allows to merge neighbouring glaciers as they grow. This change is not relevant for the manuscript or for future simulations, but it is very useful for past (paleo-) simulations where glaciers were much larger than today.

- The ice dynamics and bed inversion modules now have an option to account for lateral drag depending on the shape of the flowline's bed. The parameterization is based on Adhikari & Marshall (2012) and it is turned off per default. We will add a discussion about the sensitivity of the model to this new parametrization in Figs. 6, 8 and 10. The implementation was done by Philipp Gregor, and he will be a new co-author of the revised manuscript.

- We have worked extensively towards increasing the transparency of the model's results and reproducibility. This was done by continuing our work on model documentation, and the creation of dedicated tools for continuous monitoring of the model results. One of these tools is a benchmark tracking both the model execution time and model results (`https://cluster.klima.uni-bremen.de/~github/asv/#/`), with the goal to monitor and detect undesired changes in model behaviour. The other is a continuous monitoring and visualization of the mass-balance model comparison to individual glacier mass-balance (`https://cluster.klima.uni-bremen.de/~github/crossval`). The implementation was done by Matthias Dusch, and he is now a co-author of the revised manuscript.

- We have worked on the documentation and are continuing the work promoting the use of the model (e.g. `https://edu.oggm.org`). For his help with the documentation and the revision of this manuscript, Nicolas Champollion is now a co-author of this paper.

- A first publication based on OGGM originating from a working group outside of the "core team" is now available: Goosse et al. (2018).

- A new publication based on OGGM is now in review: Recinos et al. (2018).

- And other changes too numerous to be listed here. For the full list of model improvements, see `https://docs.oggm.org/en/latest/whats-new.html`

These changes were not all requested by the reviewers but were the result of a standard model development happening as the model gains more users. They did not change our manuscript or the model results in a significant way.

We now provide a point by point answer to the reviewers comments. ("RC" stands for "reviewer comment", "AR" for "authors response"), followed by a "diff file" of our manuscript.

**Referee 1 (David Rounce)**

**Community driven development**

You write:

*"RC: the authors state that the model is intended to be community-driven and identify many places in the manuscript where future work/modules will be developed; however, there does not appear to be much discussion of how users in the community outside of the model development team could contribute to future model development. This does not have to be discussed in detail, but I would encourage the authors to add a few lines to Section 6 about how they envision users who are not affiliated with the OGGM team to be able to contribute new modules."*

*"RC: P1 L18 - The future developments describe new physical processes and model calibration, but what about other community driven efforts that perhaps have not been thought of yet? Perhaps a phrase could be added acknowledging these other unforeseen developments."*

*"RC: P4 L2 – Perhaps future improvements and new modules? See general comment about community-driven model, but I think there could be value in adding various modules that model mass balance or glacier dynamics in different ways."*

*"P25 L33 – Here is another opportunity where perhaps you could state that the interested person could also contribute to the development of new modules as well to reinforce that this is meant to be a community-driven model? See general comment."*

**AR:** for a general discussion, see our online response (10 May 2018).

Based on this response we made the changes to the manuscript:

- we added a sentence to the abstract
- we added a sentence in the introduction
- we added a paragraph in the conclusions

**Other comments**

We addressed all editorial comments (thanks!) and do not list them here. The remaining open questions are:

*"RC: I would recommend using frontal ablation instead of calving (see general comment)."*

**AR:** Agreed. We replaced all occurrences in the text.

*"RC: P4 L23 – In the current form it appears that the mass balance model and the glacier evolution are completely separate. Is that the case or does the model compute the mass balance for a given timestep (month, year, etc.) and then allow the glacier evolution to occur? If it is the latter, then I would suggest stating how the mass balance and glacier evolution are linked in the model."*

**AR:** You are right, we did not specify this point in the manuscript. We added a paragraph to the dynamical model description in the new manuscript.

*"RC: P5 Figure 1 – Difficult to see the difference between c and d. Do c and d have a scale bar? Is this the scale bar associated with e and f? Or are the color of all the transects associated with the scale bar next to f? I assume the latter is the case, but perhaps explicitly stating this in the caption could be helpful."*

**AR:** Indeed, this figure can be hard to interpret. Figs. 1c and 1d do not have a scale bar: the colors simply serve the purpose of differentiating them. The difference between the two is subtle: in 1c, the lines are the "geometrical" ones, i.e. computed from the intersection of the flowline's normal line with the outline. In 1d, the lines widths are corrected to accurately represent the true glacier area-elevation distribution. Their representation is now centred on the flowline, which is only visible if one looks carefully (the differences between Fig 2b and 2c are easier to see, but explained later in the text).

We added an explanation to the caption and refer to the later explanation in the text.

*"RC: P6 L26 – What do you mean by "because of the lack of traceability"? Do you mean that the dates when the DEMs were acquired or the data they were generated from are unknown?"*

**AR:** yes. It can probably be tracked down, but not easily – this information is scattered around the website and probably incomplete. Visually, the DEMs look good and realistic, but biases are possible. We are aware that this is a suboptimal solution, but this really isn't in our control as long as no consistent, gap-free global DEM is available. Unfortunately, the newly released Tandem-X DEM (`https://geoservice.dlr.de/web/dataguide/tdm90/`) also seems to have many data gaps. We will continue to explore better solutions in the future.

*"RC: P7 L3 – Is there a reason for the spatial resolution of the target grid varying for each glacier, but then the Gaussian filter being applied at a constant 250 m radius? If the grid size is varied, then why is the filter size not varied as well?"*

**AR:** the DEM nominal resolution is approx. 90m (the local glacier grid is then interpolated from it). The smoothing is meant to be applied to the nominal DEM, but it is technically easier to do on the interpolated DEM. To be honest, we don't really know the influence of the smoothing radius on the results, and it might play a non-negligible role at the individual glacier scale. This is a tunable model parameter though, and users can try and test it. The change you suggest could be implemented easily, but testing its influence on the model results is the time consuming part...

We added a sentence at the appropriate location in the manuscript.

*"RC: P7 L14 – Im a bit confused by the default grid spacing. Is this still a 2d "grid" or is this now referring to a "line" spacing of the transects. Also, does map topography refer to the resolution of the DEM? Is this the same spatial resolution of the target grid (P7 L1)?"*

**AR:** Yes, the default grid spacing of "twice that of the map topography" refers to the flowline grid. The "map topography" refers to the local grid (i.e. of varying spacing depending on glacier size). As an example: Hintereisferner has a local map grid spacing of 50m, so its flowlines have a grid spacing of 100m.

We added an example and revised the semantics to avoid confusion.

*"RC: P7 L20 – Why is this necessary? Is this done to avoid sinks, to ensure that each flowline only has a single flux upstream contributing to it (although at glacier branches there would be multiple anyways), or is this done to reduce problems associated with the ice thickness inversion? Please clarify."*

**AR:** yes: the purpose of this correction is to avoid sinks ("deepenings" in the manuscript, we

will now use "sinks" instead). Sinks along a flowline are unphysical: the inversion procedure can cope with them by artificially forcing a positive slope, but this leads to results which are incompatible with the forward dynamical model, which will compensate by filling the sinks with ice before anything else. Where this happens, this would lead to undesirable spin-up issues.

We added a sentence at the appropriate location.

*"RC: P7 L26-32 – Are these the same steps as those associated with Figure 1? I would like to note that each step is much easier to visualize here in Figure 2."*

**AR**: yes, indeed. We will add a reference to Figure 2 in the caption of Figure 1.

*"RC: P9 L5 – the global precipitation correction factor is a constant of 2.5 according to Appendix A. Since it does not vary regionally, I would recommend stating its value here as well."*

**AR:** agreed.

*"RC: Does $\mu^*$ have a name?"*

**AR:** we use "temperature sensitivity" throughout the manuscript. We are not aware of any general naming convention for this parameter. In the model code and in the OGGM community, we like to call it "mu star".

We added the name in the listings of parameters after the equation.

*"RC: P10 L7 – How far away can this be? Is there a limit for how close these 10 locations need to be? Could you add the largest distances for example?"*

**AR:** no, there is no limit. The inverse distance algorithm ensures that if one glacier is very close and all the others are very far, only the close glacier will count in the weighting. However, a visual estimation based on the map in `https://oggm.org/2017/02/19/wgms-rgi-links/` leads to several thousands of kilometres distance for the Russian Arctic. This alone is an argument for the use of the OGGM-specific $t^*$, instead of the interpolation of any other "physical parameter" such as $\mu$ or similar.

We added this information and plotted the location of the WGMS reference glaciers on the global map in Appendix C for reference.

*"RC: P10 L11 – Since the residual bias is subtracted from the modelled mass balance, should this residual bias be added to equation 1? Any information on what percentage of glaciers this is required for? If its a small percentage, then perhaps its not important to include in equation 1."*

**AR:** you are right, this should be included in Eq. 1. And no, it is not negligible at all - locally, it can rise to ± 1.5 m w.e. Currently, it is not a tuning parameter (it is a residual of the calibration), but we hope to be able to use regional geodetic estimates to better constrain this value in the future.

We added a sentence at the appropriate location in the manuscript.

*"RC: P13 L22 – Can this overestimation be quantified or is it so reliant on the creep parameter that it doesnt make sense to add a range of values here? If it can be quantified (perhaps only for the default value), then it would be nice information to include."*

**AR:** for this particular idealized case (a glacier of 5km length in strong disequilibrium with its climate) the error is of about 25%. Assuming that the MB gradient and the creep parameter are known, this overestimation will depend mostly on how "strong" this disequilibrium really is. Estimating this number for real case glaciers could be an interesting study, although this effect will be very difficult to disentangle from other uncertainties.

We added a sentence at the appropriate location in the manuscript.

*"RC: P15 L15 – What does "translate in a certain h" mean? I assume it means that a new h is calculated? Furthermore, "allowing to have" also sounds awkward. Perhaps "which allows" or "which enables" various bed geometries to be present along the same flowline?"*

**AR:** yes. We followed your suggestion.

*"RC: P18 L20-22 – These sentences refer to the RGI polygons, but these polygons are partially covered in Figure 7. Is it possible to make the polygons clearer?"*

**AR:** yes, we made them more visible by changing the colorscale and the with of the glacier sections.

*"RC: P22 L28 – What is meant by strike out? Please clarify."*

**AR:** sorry, we meant that the regions with a stronger response to 21st climate change are the same in our study and that of Huss & Hock. We clarified.

**Referee 2**

**General comment**

*"RC: this paper represents a huge and ambitious undertaking, one that represents a huge leap forward in modelling global glacier response to climate change. I have nothing but admiration for the authors and their contribution. I suspect this paper will become one of the most cited contributions in GMD, as this advance in glacier modelling makes most efforts to date obsolete. As someone that has worked hard on similar kinds of modelling of individual glaciers, the effort to create a state-of- the-art, open-source model that can be applied to 10s of 1000s of glaciers on a global scale is remarkable."*

**AR:** thank you very much for your assessment and support. We would like to add that we don't think that OGGM will make other efforts "obsolete". We sincerely hope to be able to engage a wider community under the OGGM framework, making of OGGM an "umbrella" more than a model that competes with others. We are happy to see that this idea is gaining momentum, at least in the wider climate community: after a long period without funding, we are now happy to have several people working on the use and development of the model, and external groups have been successfully applying for research funds based on OGGM (University of Louvain, University of Hannover).

**Point by point answer**

*"RC: 1. The English is often incorrect or awkward, starting with the opening of the abstract: this should be "Despite their importance...." I will not make a comprehensive review of the grammatical suggestions, but a few other points are noted below."*

**AR:** thanks for your suggestions. Referee 1 also made a number of edits, and we hope to be able to get help from the Copernicus editing team before final publication.

*"RC: p.2, l.33, reference to Elmer/Ice in the ice sheet modelling community. This has also been widely applied to mountain glacier modelling - this should be noted. Also, there are other public-domain ice sheet models such as glimmer (in the CISM family) and ISSM."*

**AR:** thanks, we added these references.

*"RC: 3. p.4, l.8, this might be a suitable juncture to note SRTMs resolution ( 90 m) and the year that these elevations date to (is it 2000?). On this topic, is this reasonably well-matched with the RGI outlines? I assume the latter span a range of years - worth summarizing here."*

**AR:** yes, the outlines and topography do not always match. RGI outlines span several decades (with a large majority of outlines valid for the period 2000-2010) and the SRTM acquisition was in 2000. Since these dates change with each glacier and each DEM it is hard too summarize them all, but they are stored as attributes for each glacier during an OGGM run.

We added a discussion at the appropriate location in the manuscript.

*"RC: 4. p.6, I see that DEM resolution are summarized here, so that maybe covers off the first part of item 3. It would be worth reporting the years for each DEM though - what year(s) does the reference glacier hysometry then refer to?"*

**AR:** yes, we added the dates of each DEM dataset (when available: see our comment about DEM3 in our answer to Referee 1).

*"RC: 5. p.7, ll.1-2. I am confused by the square relation, $dx = aS^2$. Maybe I misunderstand, but should this not be a square-root relation? $dx = aS^{\frac{1}{2}}$. Dimensionally and conceptually."*

**AR:** you are right: fortunately, the error was in the manuscript, not in the code. We corrected the equation accordingly.

*"RC: 7. p.18, discussion of ice caps. Does this also refer to alpine icefields, I assume? Mountain complexes with a shared accumulation area and multiple outlets. Please clarify."*

**AR:** yes, this applies to any glacier that shares an ice divide with others (e.g. "glacier complexes" in Kienholz et al. 2013). However, we think that most glacier complexes are less problematic than ice-caps, mostly because the ice divides are more obvious in complex topography than flat ice. We clarified this point in the manuscript.

*"RC: 8. p.22, l.10 and the discussion around this. The deviations from the scaling law seem consistent with the results of Adhikari and Marshall (GRL, 2012), which was dismissed by Bahr et al (2015). Is it fair to say that the results here are consistent with the expectation that a variety of local factors such as sliding, glacier cross-sectional shape, mass balance profile, and state of disequilibrium can cause a different scaling relationship, vs. a kind of universal constant for the scaling-law exponent as argued by Bahr et al.?"*

**AR:** thanks a lot for this comment. Due to the model description nature of our manuscript, we would not like to go this path and will not make such an analysis or statement in the paper. That said, there is a lot that can be done with our global sensitivity analyses, and yes indeed the scaling law parameters change (sometimes in surprising ways) with the chosen model parametrisation. If you are interested in a follow-up study on this topic, do not hesitate to reach out to us for further discussion.

We added a reference to Adhikari and Marshall (GRL, 2012).

**References**

[revised manuscript text omitted]